# A high-content RNAi screen reveals multiple roles for long noncoding RNAs in cell division

Lovorka Stojic [1,6✉], Aaron T. L. Lun [1,7,13], Patrice Mascalchi[1,8,13], Christina Ernst[1,9], Aisling M. Redmond [1,10], Jasmin Mangei[1,11], Alexis R. Barr [2,12], Vicky Bousgouni[2], Chris Bakal [2], John C. Marioni[1,3,4], Duncan T. Odom [1,5✉] & Fanni Gergely [1✉]

Genome stability relies on proper coordination of mitosis and cytokinesis, where dynamic microtubules capture and faithfully segregate chromosomes into daughter cells. With a high-content RNAi imaging screen targeting more than 2,000 human lncRNAs, we identify numerous lncRNAs involved in key steps of cell division such as chromosome segregation, mitotic duration and cytokinesis. Here, we provide evidence that the chromatin-associated lncRNA, *linc00899*, leads to robust mitotic delay upon its depletion in multiple cell types. We perform transcriptome analysis of *linc00899*-depleted cells and identify the neuronal microtubule-binding protein, *TPPP/p25*, as a target of *linc00899*. We further show that *linc00899* binds *TPPP/p25* and suppresses its transcription. In cells depleted of *linc00899*, upregulation of *TPPP/p25* alters microtubule dynamics and delays mitosis. Overall, our comprehensive screen uncovers several lncRNAs involved in genome stability and reveals a lncRNA that controls microtubule behaviour with functional implications beyond cell division.

[1] Cancer Research UK Cambridge Institute, University of Cambridge, Li Ka Shing Centre, Robinson Way, Cambridge CB2 0RE, UK. [2] Institute of Cancer Research, 237 Fulham Road, London SW3 6JB, UK. [3] European Bioinformatics Institute, European Molecular Biology Laboratory (EMBL-EBI), Wellcome Genome Campus, Hinxton, Cambridgeshire CB10 1SD, UK. [4] Wellcome Trust Sanger Institute, Wellcome Genome Campus, Hinxton, Cambridgeshire CB10 1SA, UK. [5] Division of Regulatory Genomics and Cancer Evolution, Deutsches Krebsforschungszentrum, Im Neuenheimer Feld 280, 69120 Heidelberg, Germany. [6] Present address: Centre for Cancer Cell and Molecular Biology, Barts Cancer Institute, Queen Mary University of London, London EC1M 6BQ, UK. [7] Present address: Genentech, Inc., South San Francisco, CA, USA. [8] Present address: DRVision Technologies, Bordeaux, France. [9] Present address: School of Life Sciences, Ecole Polytechnique Fédérale de Lausanne (EPFL), 1015 Lausanne, Switzerland. [10] Present address: MRC Cancer Unit, Hutchison/MRC Research Centre, University of Cambridge, Cambridge, UK. [11] Present address: Molecular Genetics, Deutsches Krebsforschungszentrum, Im Neuenheimer Feld 280, 69120 Heidelberg, Germany. [12] Present address: MRC London Institute of Medical Sciences (LMS), Hammersmith Hospital Campus, Du Cane Road, London W12 0NN, UK. [13] These authors contributed equally: Aaron T. L. Lun, Patrice Mascalchi. ✉email: l.stojic@qmul.ac.uk; duncan.odom@cruk.cam.ac.uk; fanni.gergely@cruk.cam.ac.uk

Long noncoding RNAs (lncRNAs) are defined as RNAs longer than 200 nucleotides that lack functional open reading frames, and represent a major transcriptional output of the mammalian genome[1,2]. LncRNAs control numerous cellular processes including the cell cycle, differentiation, proliferation and apoptosis[3–5] and their deregulation is associated with human disease including cancer[6,7]. Several lncRNAs regulate the levels of key cell cycle regulators such as cyclins, cyclin-dependent kinases (CDK), CDK inhibitors and p53[8,9]. LncRNAs are also linked to cell division as they can regulate the levels of mitotic proteins[10,11] or by modulating the activity of enzymes involved in DNA replication and cohesion[12]. In addition, lncRNAs can control chromosome segregation by controlling kinetochore formation via centromeric transcription[13] or by acting as decoys for RNA-binding proteins involved in maintaining genome stability[14,15]. All of these lncRNA-dependent functions can occur through transcriptional and posttranscriptional gene regulation, chromatin organisation and/or posttranslational regulation of protein activity[5]. These mechanisms usually involve lncRNAs establishing interactions with proteins and/or nucleic acids, which allows lncRNA-containing complexes to be recruited to specific RNA or DNA targets[16]. Although lncRNAs represent >25% of all human genes (GENCODE v24), the biological significance of the majority of lncRNAs remains unknown.

Systematic screens in human cells identify protein-coding genes involved in cell survival, cell cycle progression and chromosome segregation[17–20]. Similar loss-of function screens are performed to identify lncRNAs with cell cycle functions. For example, CRISPR interference (CRISPRi) is used in high-throughput screens to identify lncRNA loci important for cell survival[21] and revealed lncRNAs whose functions were highly cell-type specific. Similar results are obtained in a CRISPR/Cas9 screen targeting lncRNA splice sites[22]. In a recent RNA interference (RNAi) screen targeting human cancer-relevant lncRNAs, Nötzold and colleagues used time-lapse microscopy imaging of HeLa Kyoto cells and identify 26 lncRNAs linked to cell cycle regulation and cell morphology[23]. However, this screen only studied ~600 lncRNAs in the genome with respect to a limited number of phenotypes.

With an aim to identify lncRNAs with functions in cell division, we perform a more comprehensive high-content imaging RNAi screen involving the depletion of 2231 lncRNAs in HeLa cells. We develop image analysis pipelines to quantify a diverse set of mitotic features in fixed cells, and discover multiple lncRNAs with roles in mitotic progression, chromosome segregation, and cytokinesis. We focus on *linc00899*, a hitherto uncharacterised lncRNA that regulates mitotic progression by repressing the transcription of the microtubule-stabilising protein *TPPP/p25*. Our study demonstrates the regulatory function of *linc00899* in mitotic microtubule behaviour and provides a comprehensive imaging data resource for further investigation of the roles of lncRNAs in cell division.

## Results

**High-content RNAi screen identifies lncRNAs in cell division.** To identify lncRNAs involved in regulating cell division, we performed two consecutive RNAi screens (screen A and B). Briefly, we transfected HeLa cells with the human Lincode small interfering RNA (siRNA) library targeting 2231 lncRNAs (Fig. 1a; Supplementary Data 1) and examined their effects using high-content screening of mitotic phenotypes. Each lncRNA was targeted with a SMARTpool of four different siRNAs. Following 48-h incubation, cells were fixed and processed for immunostaining and subsequent automated image acquisition and analysis. In screen A, antibodies targeting CEP215 (to label centrosomes),

α-tubulin (to label the microtubule cytoskeleton), phalloidin (to label the actin cytoskeleton) and Hoechst (to label nuclei) were used. In screen B (Fig. 1b–d), phospho-histone H3 (PHH3; to specifically label mitotic cells), α-tubulin, γ-tubulin (to label centrosomes) and Hoechst was used. We used these two screens as independent approaches to robustly identify lncRNAs with functions in mitotic progression, chromosome segregation and cytokinesis.

In each screen, we employed automated image analysis to segment the cells and developed in-house pipelines to quantify defects in each of abovementioned categories upon lncRNA depletion (Supplementary Fig. 1a). First, for defects in mitotic progression, we determined the percentage of mitotic cells (also called mitotic index), because an increase in the mitotic index implies a delay or block in mitotic progression. We performed nuclear segmentation and computed the mitotic index (Supplementary Fig. 1b) where mitotic cells were identified by the presence of mitotic spindle staining (detected by α-tubulin and CEP215) in screen A or by positive PHH3 staining of chromosomes in screen B. Second, for quantification of chromosome segregation defects, a category that includes chromatin bridges and lagging chromatids, we identified anaphase cells based on α-tubulin staining between the separating nuclei, in addition to Hoechst (DNA) signal (Supplementary Fig. 1c). Third, to evaluate defects in the execution of cytokinesis, we segmented the cytoplasm of interphase cells and scored the number of cells with cytokinetic bridges based on α-tubulin staining (Supplementary Fig. 1d).

For each category, we ascertained lncRNAs for which depletion increased the frequency of defects relative to the mean across all lncRNAs in the siRNA library. Negative control siRNA and cells without any siRNA treatment were used as controls, for which we observed no systematic differences in the frequencies of each phenotype (Fig. 1b–d). We identified candidate lncRNAs involved in mitotic progression (Fig. 1a; *linc00899* and *C1QTNF1-AS1*), chromosome segregation (Fig. 1b; *PP7080* and *linc00883*) and cytokinesis (Fig. 1c; *linc00840* and *loc729970*). As a positive control for mitotic progression defects, we used a SMARTpool against the protein-coding gene *Ch-TOG/CKAP5*, whose depletion leads to mitotic delay and increased mitotic index[24] (Fig. 1b). For chromosome segregation, we successfully identified the lncRNA *NORAD* (Fig. 1c), depletion of which increases the rate of chromosome segregation errors[14,15]. Supplementary Data 2 contains raw data and computed Z-scores for each lncRNA and phenotype.

To confirm our findings, we conducted a validation screen targeting the top 25 lncRNA candidates identified in the initial screens for mitotic progression and cytokinesis (Supplementary Data 3). Depletion of each lncRNA was performed in two biological (and in total eight technical) replicates. For mitotic progression, this screen corroborated the increase in mitotic index following depletion of *linc00899* and *C1QTNF1-AS1* (Supplementary Fig. 2a). Although *loc100289019*-depleted cells also displayed an elevated mitotic index, levels of the lncRNA did not change upon RNAi[25]. For the cytokinesis category, we observed an increase in the number of cells with cytokinetic bridges after *linc00840* depletion and a decrease after *loc729970* depletion, but neither led to multinucleation (Supplementary Fig. 2b, c). Furthermore, elevated mitotic index and cytokinesis defects were not associated with reduced cell viability for these lncRNAs (Supplementary Fig. 2d). As positive controls, we used *Ch-TOG* and *ECT2* (a key regulator of cytokinesis)[26], the depletion of which led to expected phenotypes: an increased number of mitotic and multinucleated cells, respectively (Supplementary Fig. 2a–c).

Mitotic perturbations caused by depletion of the lncRNA candidates were further characterised by time-lapse microscopy

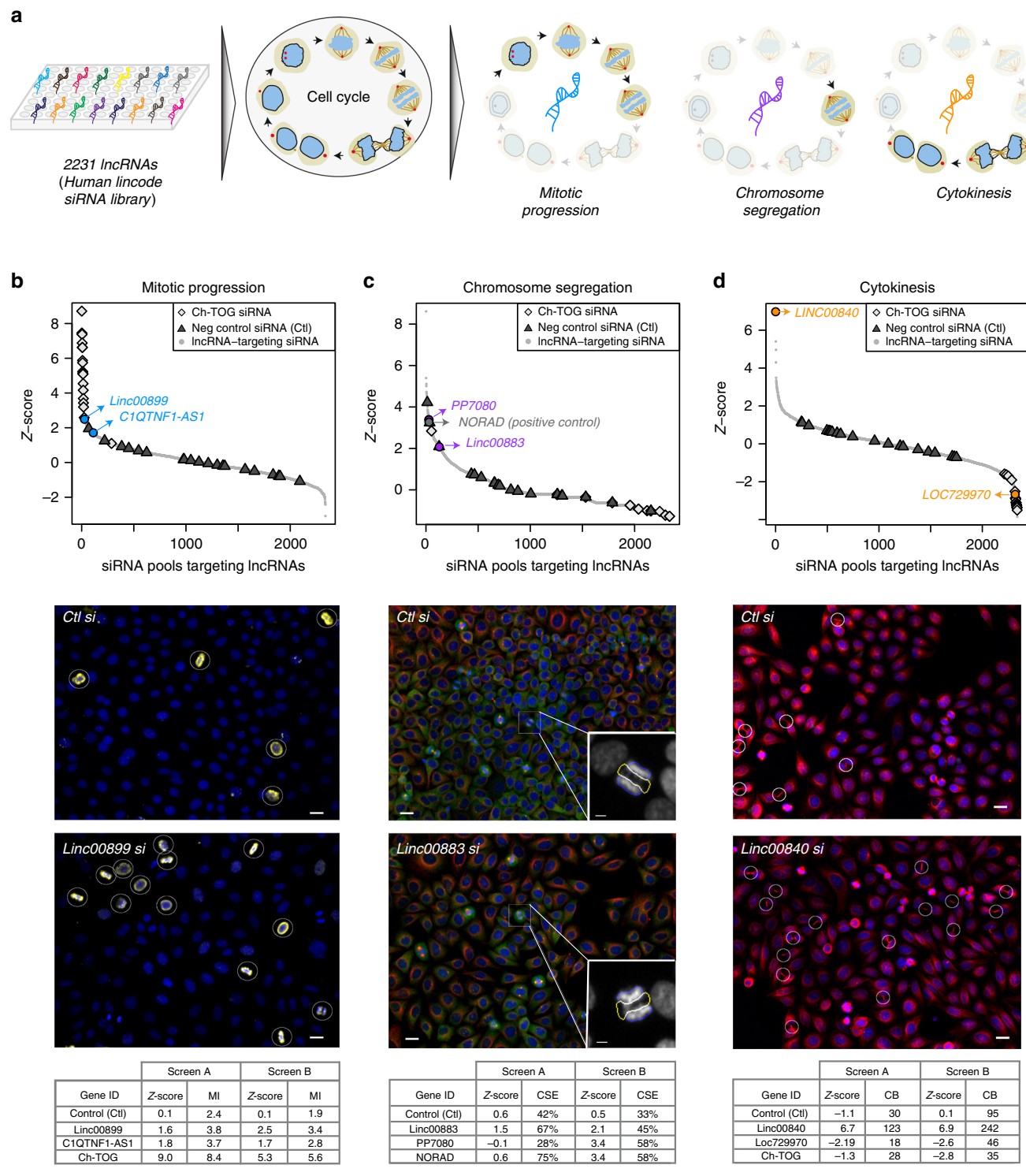

imaging to investigate the dynamics of each phenotype. As expected, a marked mitotic delay was observed in HeLa cells depleted of *linc00899* and *C1QTNF1-AS1*, lncRNAs associated with increased mitotic index (Supplementary Fig. 3). Next, we depleted *linc00883* and *PP7080*, lncRNAs with potential functions in chromosome segregation (Supplementary Fig. 4). Using time-lapse microscopy imaging of HeLa Kyoto cells stably expressing histone H2B-mCherry (a chromatin marker) and eGFP-α-tubulin (a microtubule marker)[18], we found that depletion of *linc00883* and *PP7080* increased the rate of chromosome segregation errors to a similar extent as that of *NORAD*. We then depleted *linc00840*

and *loc729970* (Supplementary Fig. 5), lncRNAs from the cytokinesis category, and found that knockdown of *linc00840* doubled the time required for cells to cleave the cytokinetic bridge, whereas knockdown of *loc729970* resulted in shorter cytokinesis. Overall, our screen identified functions of lncRNAs in the control of cell division, supporting the idea that lncRNAs play an important role in cell cycle progression.

**Molecular characterisation of *linc00899* and *C1QTNF1-AS1*.** Two lncRNAs associated with delayed mitotic progression,

**Fig. 1 Identification of lncRNAs involved in regulation of cell division. a** Schematic representation of the high-throughput RNAi imaging screen for lncRNAs regulating three mitotic processes: mitotic progression, chromosome segregation and cytokinesis. The screen depleted each of 2231 lncRNAs in HeLa cells using the Human Lincode siRNA library (Dharmacon). **b** Z-scores for mitotic progression defects upon depletion of each lncRNA in the siRNA library. Each point corresponds to a single lncRNA where the Z-score was computed based on the mean mitotic index (MI). siRNAs against the protein-coding gene *Ch-TOG* were used as positive controls, in addition to negative control siRNAs (Ctl, from Ambion). Representative images from the top candidate (*linc00899*, in blue) are also shown, with PHH3 in yellow indicating mitotic cells (white circles). **c** Z-scores for chromosome segregation defects upon lncRNA depletion, similar to **b**. The Z-score per lncRNA was computed from the mean number of chromosome segregation errors (CSE). Here *NORAD* (grey) was used as a positive control. Top candidates are highlighted in purple. Representative images from one of the top candidates (*linc00883*, in purple) are also shown with staining for α-tubulin (red), PHH3 (green) and γ-tubulin (yellow). Inset depicts normal anaphase cell (blue area) or anaphase cell with CSE (yellow area). **d** Z-scores for cytokinesis defects upon lncRNA depletion, similar to **a**. The Z-score for each lncRNA was computed based on the mean number of cells with cytokinetic bridges (CB). Representative images from the top candidate (*linc00840*, in orange) are shown with staining for α-tubulin (red) and DNA (blue). CB are depicted in white circles. All Z-scores shown here are from screen B. Some of the top candidates are shown in colour and labelled in each plot. The scale bar for all images is 20 μm. Tables below each panel represent the raw data and calculated Z-scores for top lncRNA candidates for each category from two independent screens (A and B).

*linc00899* and *C1QTNF1-AS1*, were selected for in-depth functional analysis. *Linc00899* and *C1QTNF1-AS1* are spliced and polyadenylated lncRNAs. *Linc00899* (also known as *loc100271722* or *ENSG00000231711*) is a multi-exonic intergenic lncRNA located on chromosome 22 and is ~1.6 kb long. *C1QTNF1-AS1* (also known as *ENSG00000265096*) is an lncRNA on chromosome 17 that is ~1 kb long and is antisense to a protein-coding gene "C1q And TNF Related 1" (C1QTNF1/CTRP1). Both lncRNAs are annotated in GENCODE, show signs of active transcription (Fig. 2a) and have low protein-coding potential (Supplementary Fig. 6a). Although we did not identify a syntenic ortholog for *linc00899* in the mouse genome, short stretches of conserved regions[27] are present within exon 1 (Supplementary Fig. 6b). This places *linc00899* in a group of lncRNAs with conserved exonic sequences embedded in a rapidly evolving transcript architecture[28]. Based on the syntenic position of protein-coding gene *C1QTNF1*, we found a mouse ortholog for *C1QTNF1-AS1* (*GM11747*, *ENSMUG000000086514*) that is also antisense to the mouse *C1qtnf1*. Thus *C1QTNF1-AS1* is an lncRNA that is conserved across mouse and human, while *linc00899* contains short conserved stretches at its 5′ end representing possible functional domains[29,30].

We then validated the expression of both lncRNAs in HeLa cells using a variety of techniques. cDNA generated from polyadenylated RNA was used for rapid amplification of cDNA ends (RACE) to define the locations of the 5′ cap and 3′ end identifying several isoforms for *linc00899* and *C1QTNF1-AS1* (Supplementary Fig. 6c). The diversity of isoforms for both lncRNAs is consistent with a previous study on lncRNA annotation[27]. Expression data from ENCODE cell lines indicated that *linc00899* and *C1QTNF1-AS1* were present in both the nucleus and cytoplasm of HeLa-S3 cells (Fig. 2b), which we further confirmed by RNA fluorescence in situ hybridisation (RNA FISH) (Fig. 2c). Quantitative real-time polymerase chain reaction (qPCR) analyses on RNA extracted from different cellular fractions revealed that *linc00899* but not *C1QTNF1-AS1* is associated with chromatin (Fig. 2d). Although some *linc00899* foci remain detectable in mitotic cells (Supplementary Fig. 6d), they do not associate with mitotic chromatin, arguing against a mitotic bookmarking role for these lncRNAs. *linc00899* and *C1QTNF1-AS1* are estimated to occur, on average, in five and two copies per cell, respectively, and do not show cell cycle dependency in their expression (Supplementary Fig. 6e, f). In the Genotype-Tissue Expression (GTEx) RNA-seq dataset[31] from human tissue, *C1QTNF1-AS1* was highly expressed in the adrenal gland and spleen, while *linc00899* was broadly expressed in most of the tissues, with uterus being the highest (Supplementary Fig. 6g).

**Linc0889 and C1QTNF1-AS1 facilitate timely mitotic progression.** To characterise the mitotic phenotype further, we depleted

*linc00899* and *C1QTNF1-AS1* in HeLa and HeLa Kyoto cells and examined the effect on mitotic progression with immuno-fluorescence and time-lapse microscopy imaging, respectively. Quantification of lncRNA-depleted cells using RNAi revealed an increase in the mitotic index compared to cells treated with control siRNA (Fig. 3a, b). This was confirmed by time-lapse microscopy imaging of HeLa Kyoto cells where depletion of *linc00899* and *C1QTNF1-AS1* resulted in increased mitotic duration (Supplementary Fig. 7a–c). We found that cells treated with control siRNA initiated anaphase onset at 40 ± 10 min (median ± SD), whereas the cells depleted of *linc00899* and *C1QTNF1-AS1* initiate anaphase onset at 100 ± 173 and 240 ± 146 min, respectively (Fig. 3c). Although unaligned chromosomes near spindle poles were detectable in a small population of *linc00899*-depleted cells (and these showed delays of >4 h), in the majority of cells bipolar spindle formation and chromosome congression occurred with normal kinetics, and cells exhibited a delay in the metaphase to anaphase transition (Fig. 3d upper panels and Supplementary Movies 1 and 2). By contrast, nearly all *C1QTNF1-AS1*-depleted cells showed an impairment of chromosome congression to the metaphase plate (Fig. 3d lower panels and Supplementary Movie 3).

We next asked whether these phenotypes could be recapitulated by a loss-of-function (LOF) method other than RNAi. We used locked nucleic acid (LNA) oligonucleotides as a complementary method to target lncRNA expression due to their higher efficiency in depleting nuclear compared to cytoplasmic lncRNAs[32,33]. We successfully depleted both *linc00899* and *C1QTNF1-AS1* with two different LNAs and selected the LNA with more efficient depletion (denoted LNA1) for subsequent studies (Fig. 3e). Similar to RNAi, we observed an increase in the mitotic index following LNA1-mediated depletion of *linc00899* and *C1QTNF1-AS1* (Fig. 3f) and confirmed the mitotic delay with time-lapse microscopy imaging of HeLa Kyoto cells (Fig. 3g, h; Supplementary Fig. 7d–f and Supplementary Movies 4–6). The apparent increase in α-tubulin intensity at the spindle poles of *C1QTNF1-AS1*-depleted cells is most likely due to spindles being short and multipolar. Together, these data indicate that *linc00899* and *C1QTNF1-AS1* have biological functions in regulation of mitotic progression, albeit through different mechanisms.

To further confirm the phenotype of *C1QTNF1-AS1* depletion, we inserted a polyadenylation poly(A) signal (pAS)[34] downstream of the *C1QTNF1-AS1* transcriptional start site (TSS) using CRISPR-Cas9 gene editing. This method allows lncRNAs to be transcribed but terminates them prematurely due to the inserted pAS, preventing the expression of full-length lncRNA transcripts. We obtained four homozygous clones for *C1QTNF1-AS1* with similar knockdown efficiency, all of which displayed mitotic delay similar to

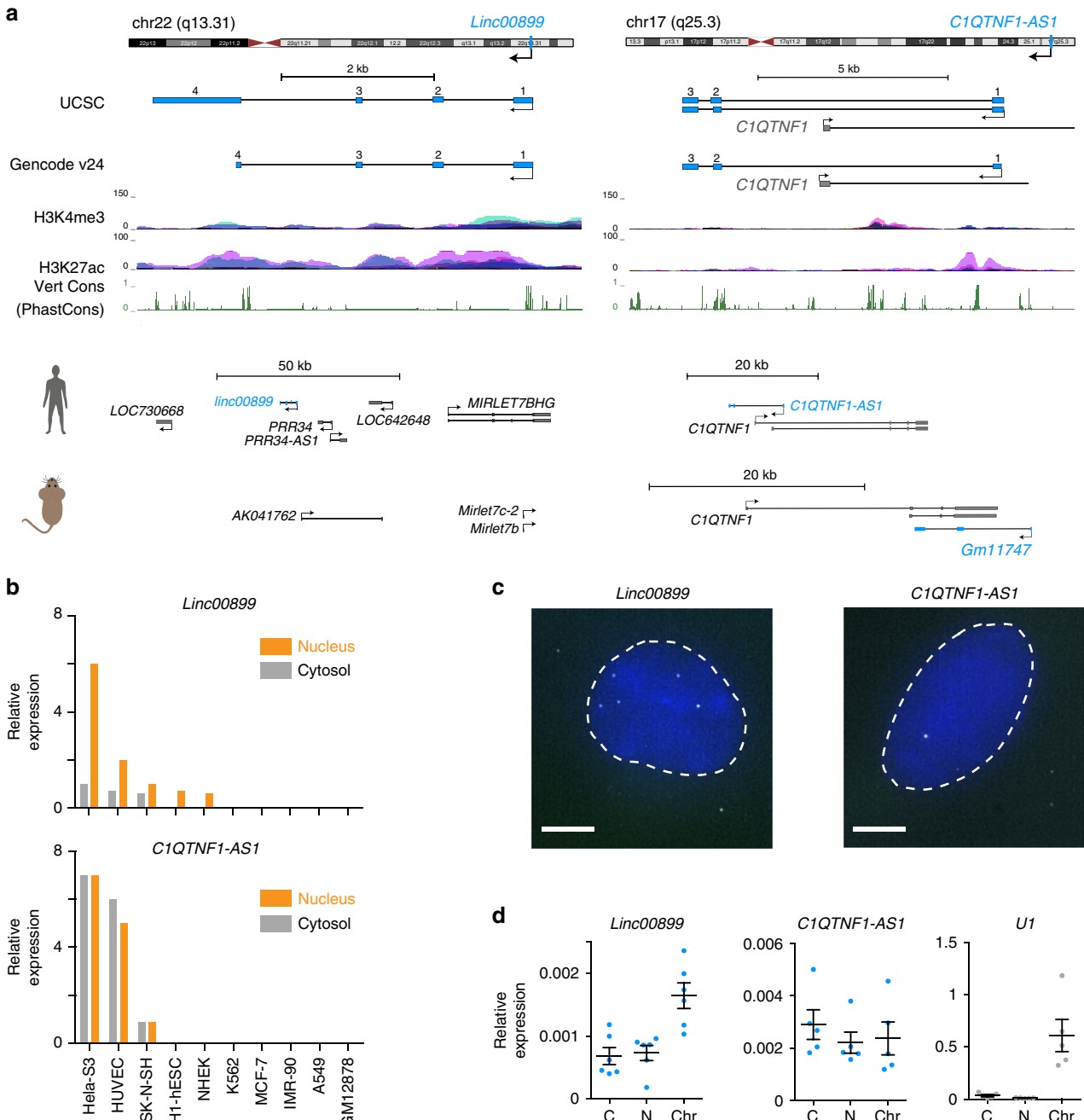

**Fig. 2 Molecular characterisation of the *linc00899* and *C1QTNF1-AS1* lncRNAs. a** Schematic representation of the genomic landscape surrounding *linc00899* (annotated in RefSeq as NR_027036; Gencode gene *ENSG00000231711*; chr22:46039907-46044868, hg38) and *C1QTNF1-AS1* (annotated in RefSeq as NR_040018/NR_040019; Gencode gene *ENSG00000265096*; chr17:79019209-79027601, hg38). Marks of active transcription in HeLa cells (H3K4me3 and H3K27ac, obtained from ENCODE via the UCSC browser) and conservation scores by PhastCons are also shown. Putative conserved mouse lncRNAs based on syntenic conservation are shown below. **b** Expression of *linc00899* and *C1QTNF1-AS1* in the nucleus (orange) and cytosol (grey) of ENCODE cell lines, shown as reads per kilobase of exon per million reads mapped. **c** Single-molecule RNA FISH using exonic probes (white) against *linc00899* and *C1QTNF1-AS1*. Nuclei were stained with DAPI (blue). Scale bar, 5 μm. **d** qPCR quantification of lncRNA levels in RNA fractions extracted from different cellular compartments (C cytoplasm, N nucleoplasm, Chr chromatin). *U1* small nuclear RNA was used as a positive control for the chromatin fraction. Error bars represent the standard error of the mean ± S.E.M from at least five independent experiments. Source data are provided as a Source Data file.

RNAi- and LNA-mediated depletion of *C1QTNF1-AS1* (Supplementary Fig. 8). These results suggest that the *C1QTNF1-AS1* transcript, and not transcription at the *C1QTNF1-AS1* locus, is required for the regulation of mitotic progression. Similar experiments for *linc00899* were hindered by the presence of over four copies of *linc00899* in the HeLa genome (https://cansar.icr.ac.uk/cansar/cell-lines/HELA/copy_number_variation/) preventing the

generation of homozygous clones despite the use of two different guide RNAs (gRNAs) targeting different regions of *linc00899*.

## *Linc0889*-dependent regulation of *TPPP* in mitotic progression.
To reveal transcriptional regulatory functions of *linc00899* and *C1QTNF1-AS1*, we performed RNA sequencing (RNA-seq) of

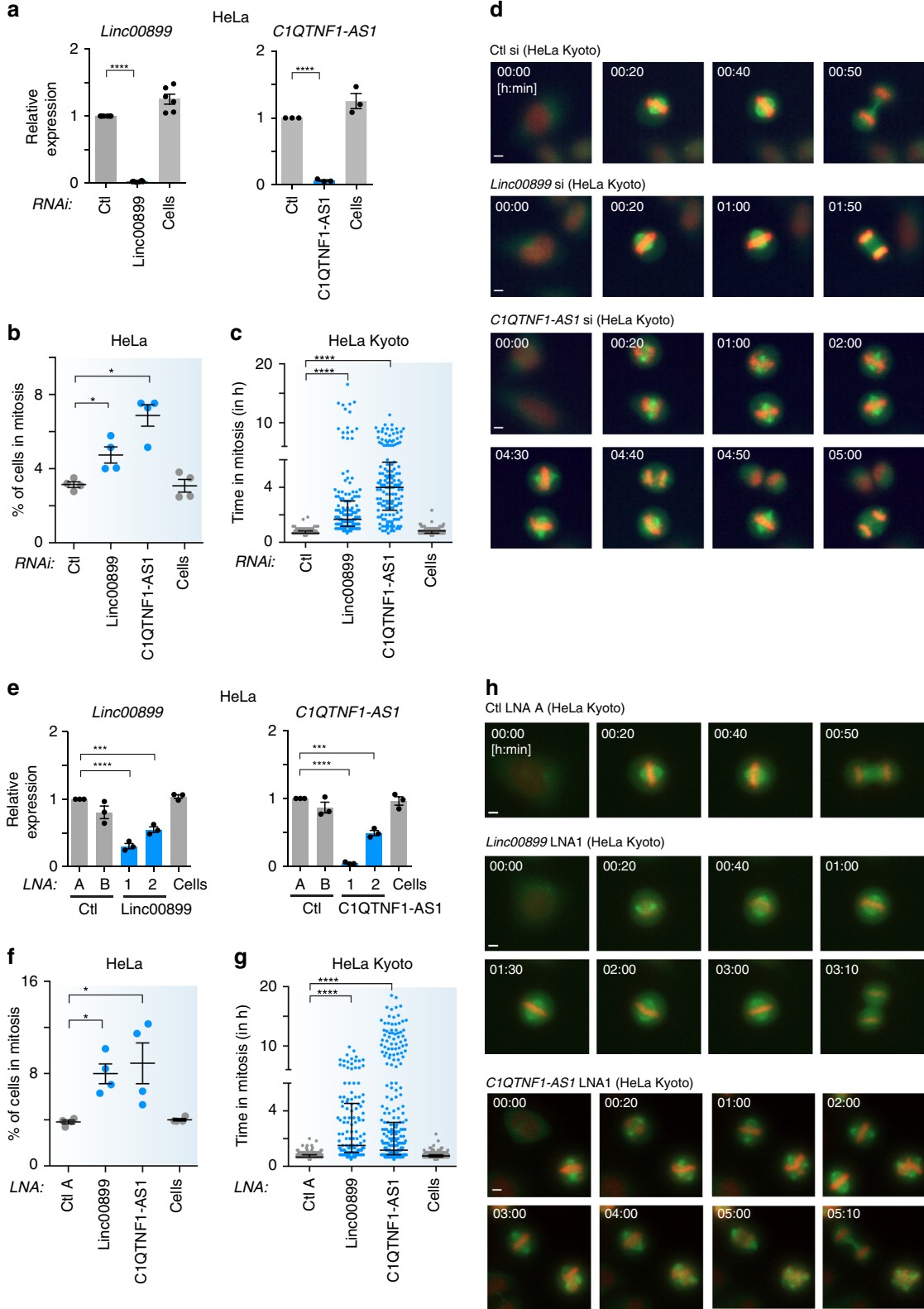

cells following RNAi- and LNA-mediated depletion of each lncRNA. We selected the subset of differentially expressed genes (DEGs) that changed in a consistent direction with both LOF methods to minimise any potential method specific off-target effects as shown previously[25]. For *C1QTNF1-AS1*, a single DEG was detected, which was *C1QTNF1-AS1* itself (Fig. 4a). These data argue against a transcriptional

role for *C1QTNF1-AS1* and suggest that its function in mitotic progression could depend on *C1QTNF1-AS1* protein interactors.

For *linc00899*, we identified eight DEGs in common across the two LOF methods (Fig. 4a), of which four were changing in the same direction with both methods (Fig. 4b). Since we were not able to validate changes in *RAI14*, *DNAAF5* and *ITGB1BP1*

**Fig. 3 Linc00899 and C1QTNF1-AS1 regulate mitosis through different mechanisms. a** Expression of *linc00899* and *C1QTNF1-AS1* after RNAi depletion, as measured by qPCR. Results are also shown for negative control siRNAs (Ctl, from Ambion) and for cells treated with transfection reagent only (Cells). $n =$ 3 (*C1QTNF1-AS1*) and 6 (*linc00899*) biological replicates, ****$P < 0.0001$ by two-tailed Student's *t* test. Expression values are presented relative to control siRNA. **b** Changes in the mitotic index (MI, based on PHH3 and Hoechst staining) after RNAi-mediated depletion of *linc00899* and *C1QTNF1-AS1* determined 48 h after siRNA transfection. $n = 4$ biological replicates, *$P < 0.05$ by Mann–Whitney test. **c** Quantification of mitotic duration from time-lapse microscopy imaging after depletion of *linc00899* and *C1QTNF1-AS1* in HeLa Kyoto cells. We analysed $n = 183$ for cells treated with Ctl si, $n = 241$ for Cells, $n = 163$ for *linc00899* siRNAs and $n = 147$ for *C1QTNF1-AS1* siRNAs. Bars show the median and the interquartile range from three biological replicates. ****$P < 0.0001$ by Mann–Whitney test. **d** Representative still images from time-lapse microscopy from **c**. Scale bar, 5 μm. **e** Expression of *linc00899* and *C1QTNF1-AS1* after depletion using two different LNA gapmers, as measured by qPCR. Results are also shown for negative control LNAs (Ctl LNA A and B) and for Cells. $n = 3$ biological replicates, ***$P < 0.001$ and ****$P < 0.0001$ by two-tailed Student's *t* test. Expression levels are presented relative to Ctl LNA A. **f** Changes in the MI after depletion of *linc00899* and *C1QTNF1-AS1* using LNA1 gapmers, as measured in **b**. $n = 4$ biological replicates, *$P < 0.05$ by Mann–Whitney test. **g** Quantification of mitotic duration as in **c** after depleting *linc00899* and *C1QTNF1-AS1* with LNA1 gapmers. We analysed $n = 273$ for Ctl LNA A, $n = 312$ for Cells, $n = 138$ for *linc00899* LNA1 and $n = 349$ for *C1QTNF1-AS1* LNA1. Bars show the median and interquartile range from three biological replicates. ****$P < 0.0001$ by Mann–Whitney test. **h** Representative still images from time-lapse microscopy from **g**. Scale bar, 5 μm. Data are shown as mean ± S.E.M for **a**, **b**, **e**, **f**. Mitotic duration in **c**, **g** was defined from NEBD ($t = 0$ min) to anaphase onset. Source data are provided as a Source Data file.

expression at the mRNA and protein levels (Supplementary Fig. 9), we decided to focus on *TPPP/p25*.

*TPPP/p25* is a tubulin polymerisation-promoting protein with established roles in microtubule dynamics and mitosis[35,36]. *TPPP* was upregulated upon RNAi- and LNA-mediated depletion of *linc00899* (Fig. 4c, d), which we validated at the protein level in asynchronous (Fig. 4e, f) as well as in mitotic cells (Supplementary Fig. 10a).

To address whether the mitotic delay in *linc00899*-depleted cells is due to increased *TPPP* levels, we performed single and double knockdowns of *linc00899* and *TPPP* and analysed the mitotic progression (Fig. 4g). Whereas *TPPP* knockdown alone did not affect mitotic timing, depletion of *linc00899* alone in HeLa cells led to a mitotic delay, consistent with results from HeLa Kyoto cells (Fig. 3c, g). However, cells with simultaneous depletion of *linc00899* and *TPPP* progressed through mitosis with near-normal timing, a phenotype observed with both LOF methods. We confirmed that co-depletion of *linc00899* and *TPPP* rescued the previously observed upregulation of the latter with qPCR (Fig. 4h, i). Similar results were obtained with an additional LNA oligonucleotide targeting the first intron of *linc00899* (Supplementary Fig. 10b, c). These data suggest that *linc00899* needs to be depleted at least by 50% to attain *TPPP* upregulation and mitotic delay in HeLa cells (Supplementary Fig. 10d, e).

**Linc00889 controls microtubule dynamics in the spindle.** *TPPP* is known to localise to the mitotic spindle, and its overexpression influences microtubule dynamics and stability in mammalian cells[35–38]. Overexpression of *TPPP* increases tubulin acetylation and also microtubule stability via microtubule bundling[36]. Thus we tested whether depletion of *linc00899*, which leads to upregulation of *TPPP*, had a similar effect. Immunofluorescence of *linc00899*-depleted cells using antibodies against α-tubulin (a microtubule marker) and acetylated α-tubulin (a marker of long-lived microtubules)[39] showed a marked increase in acetylated α-tubulin levels (Fig. 5a), consistent with *linc00899*-depleted cells containing more long-lived microtubules. We next benchmarked the effects of *linc00899* depletion on microtubules to those exerted by paclitaxel (taxol), a microtubule-stabilising agent that suppresses microtubule dynamics at nanomolar doses[40,41]. Cells were treated with 0.5–3 nM taxol, concentrations that suppress microtubule dynamics without affecting spindle morphology[42] because high doses of taxol (5 nM–1 μM) block microtubule depolymerisation leading to highly aberrant mitotic spindles and cell death[40]. We analysed cells after 1- or 20-h taxol treatment. We found that the impact of *linc008999* depletion on acetylated α-tubulin levels was comparable to that of 3 nM taxol (Fig. 5b, c),

suggestive of impaired microtubule dynamics in *linc008999*-depleted cells.

As *TPPP* influences the growth velocity of microtubules by affecting their stability[36], we examined the localisation of EB1 protein, which specifically associates with growing microtubule plus-ends where it regulates microtubule dynamics[43]. As expected, EB1 staining was apparent at the spindle pole and throughout the spindle in control cells but it was much reduced in intensity upon *linc00899* depletion (Fig. 5d). Importantly, reduced EB1 signal was not due to diminished microtubule levels, because α-tubulin staining of the mitotic spindle was comparable between *linc00899*-depleted and control cells. Quantification of EB1 signal in mitotic cells confirmed the decrease in EB1 levels, indicating that *linc00899* depletion lessens the number of growing microtubule ends, a phenotype consistent with a reduction in microtubule dynamics (Fig. 5e). Taken together, these data demonstrate that, by controlling *TPPP* expression levels, *linc00899* limits the number of long-lived microtubules and maintains normal microtubule dynamics in mitotic HeLa cells.

To gain insight into why depletion of *linc00899* and *C1QTNF1-AS1* causes a mitotic delay, we examined the distribution of the kinetochore (CREST) and spindle (α-tubulin) markers in HeLa cells depleted of these lncRNAs (Fig. 6a). In contrast to controls, where most chromosomes were present in a narrow or wide metaphase plate, ~20% of *linc00899*-depleted mitotic cells displayed narrow metaphase plates with single unattached kinetochore pairs near the poles (congression defect I; Fig. 6a and Supplementary Fig. 11). Congression defects appeared even more severe in *C1QTNF1-AS1*-depleted cells, which showed very wide metaphase plates with several clusters of chromosomes surrounding both spindle poles (congression defect II; Fig. 6a and Supplementary Fig. 11). Phenotypes were confirmed with LNA-mediated depletion for both lncRNAs. The differences in mitotic phenotypes suggest that the mechanisms through which these two lncRNAs control mitotic progression are likely to be different.

Spindle assembly checkpoint (SAC) signalling at the kinetochores ensures that each and every chromosome is bi-oriented on the metaphase plate before anaphase is initiated[44]. To assess the status of SAC, we immunostained control and *linc00899*-depleted cells for the presence of Mad2, an essential component of the SAC signalling machinery[45,46] (Fig. 6b). Frequency of Mad2 on narrow metaphase plates appeared slightly reduced in *linc00899*-depleted cells when compared to controls (12% vs 20%). However, this may not be of biological significance because 89% of *linc00899*-depleted cells with narrow metaphase plates exhibited Mad2 signal on the sister kinetochores of these uncongressed chromosomes. Thus active SAC

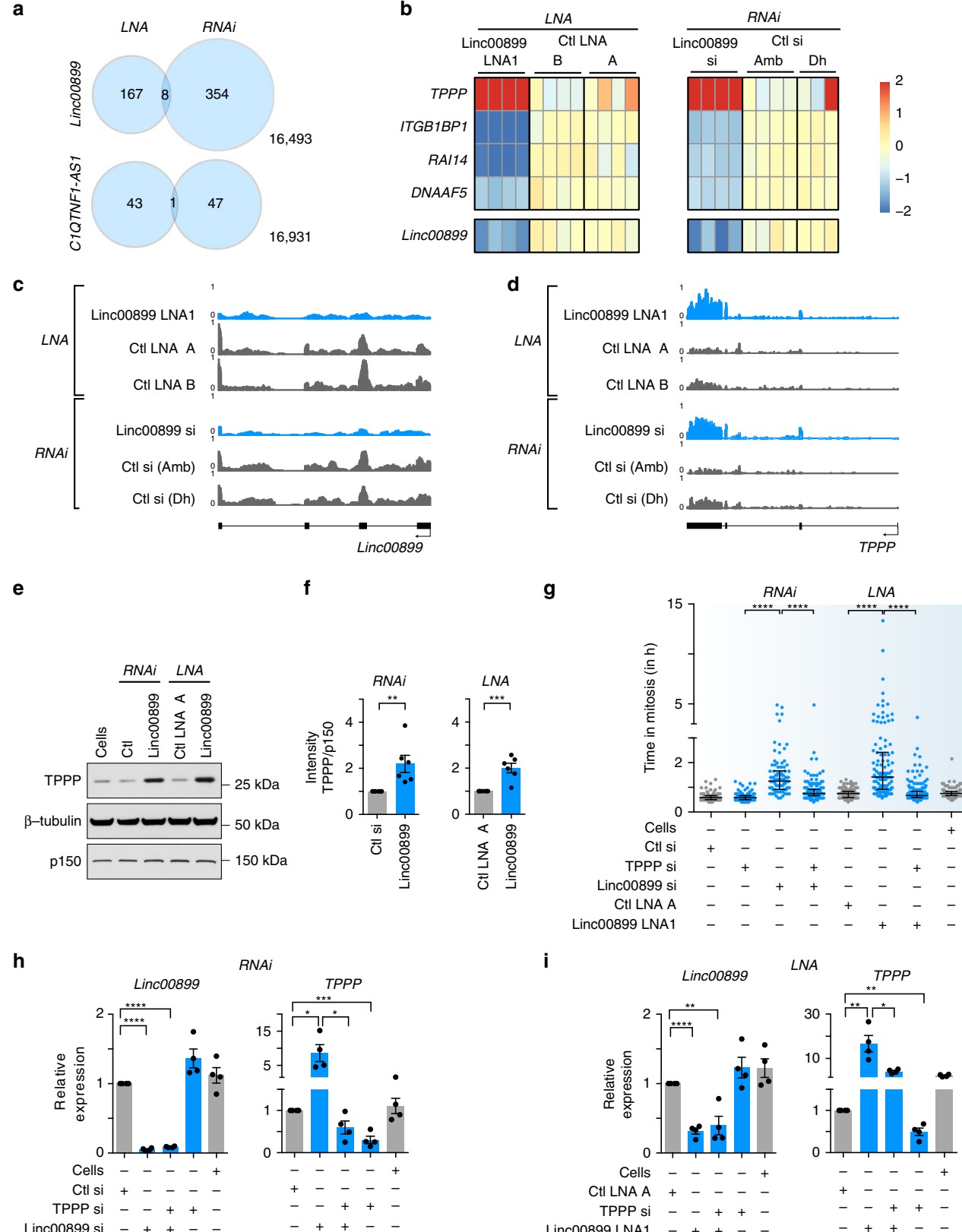

due to uncongressed chromosomes contributes to the mitotic delay in *linc00899*-depleted cells.

Once chromosome bi-orient, they come under tension from pulling forces generated by kinetochore microtubules. Given the central role of microtubule dynamics in this process, and its impairment in *linc00899*-depleted cells, we determined the interkinetochore distances (inter-KT) of chromosomes, which are known to report of the tension that kinetochore pairs are subjected to (Fig. 6c, d)[47]. Whereas in cells treated with negative control siRNA or control LNA, the mean inter-KT distance was $1.4 \pm 0.2 \, \mu m$, bi-oriented chromosomes in *linc00899*-depleted cells had a reduced mean inter-KT distance of $1.2 \pm 0.2 \, \mu m$

**Fig. 4 *TPPP*, a microtubule-stabilising protein, is a target of *linc00899* in regulation of mitosis. a** Venn diagram of DEGs detected by RNA-seq after depletion of *linc00899* and *C1QTNF1-AS1* using RNAi and LNA1 gapmers. The total number of genes was 17022. **b** Heat map of DEGs detected with RNA-seq after depletion of *linc00899* using RNAi and LNA gapmers. Three to four biological replicates were generated for each condition and compared to appropriate negative controls (Ctl)—A and B for LNA gapmers, Ambion (Amb) and Dharmacon (Dh) for RNAi. Only DEGs changing in the same direction with both LOF methods are shown. *Linc00899* itself is shown as a reference. **c** Genome tracks of RNA-seq coverage for *linc00899* before and after depletion of *linc00899* using RNAi or LNA1 gapmers. The tracks were constructed from averages of 3–4 biological replicates for each condition. **d** Genome tracks of RNA-seq coverage for *TPPP*, as in **c**. **e** Representative western blot of TPPP levels after depletion of *linc00899*. β-Tubulin and p150 were used as two loading controls. **f** Densitometric analysis of TPPP levels in **e**. $n = 6$ biological replicates. **$P < 0.01$ and ***$P < 0.001$ by two-tailed Student's *t* test. **g** Quantification of mitotic duration of HeLa cells after single and double knockdown of *linc00899* and *TPPP* using RNAi or LNAs. We analysed $n = 153$ for cells treated with transfection reagent alone (Cells), $n = 187$ for negative control siRNA (Ctl, from Ambion), $n = 207$ for *TPPP* siRNAs, $n = 97$ for *linc00899* siRNAs, $n = 204$ for *linc00899* and *TPPP* siRNAs, $n = 182$ for Ctl LNA A, $n = 115$ for *linc00899* LNA1 and $n = 246$ for *linc00899* LNA1 and *TPPP* siRNA. For each condition, we show the median with interquartile range from two biological replicates. ****$P < 0.0001$ by Mann–Whitney test. Mitotic duration was defined from NEBD ($t = 0$ min) to anaphase onset. **h–i** Expression of *linc00899* and *TPPP* after single or double knockdown with RNAi (**h**) or LNA1 gapmers (**i**) to deplete *linc000899*. $n = 4$ biological replicates, *$P < 0.05$, **$P < 0.01$, ***$P < 0.001$ and ****$P < 0.0001$ by two-tailed Student's *t* test. Data are shown as mean ± S.E.M. for **f**, **h**, **i**. Source data are provided as a Source Data file.

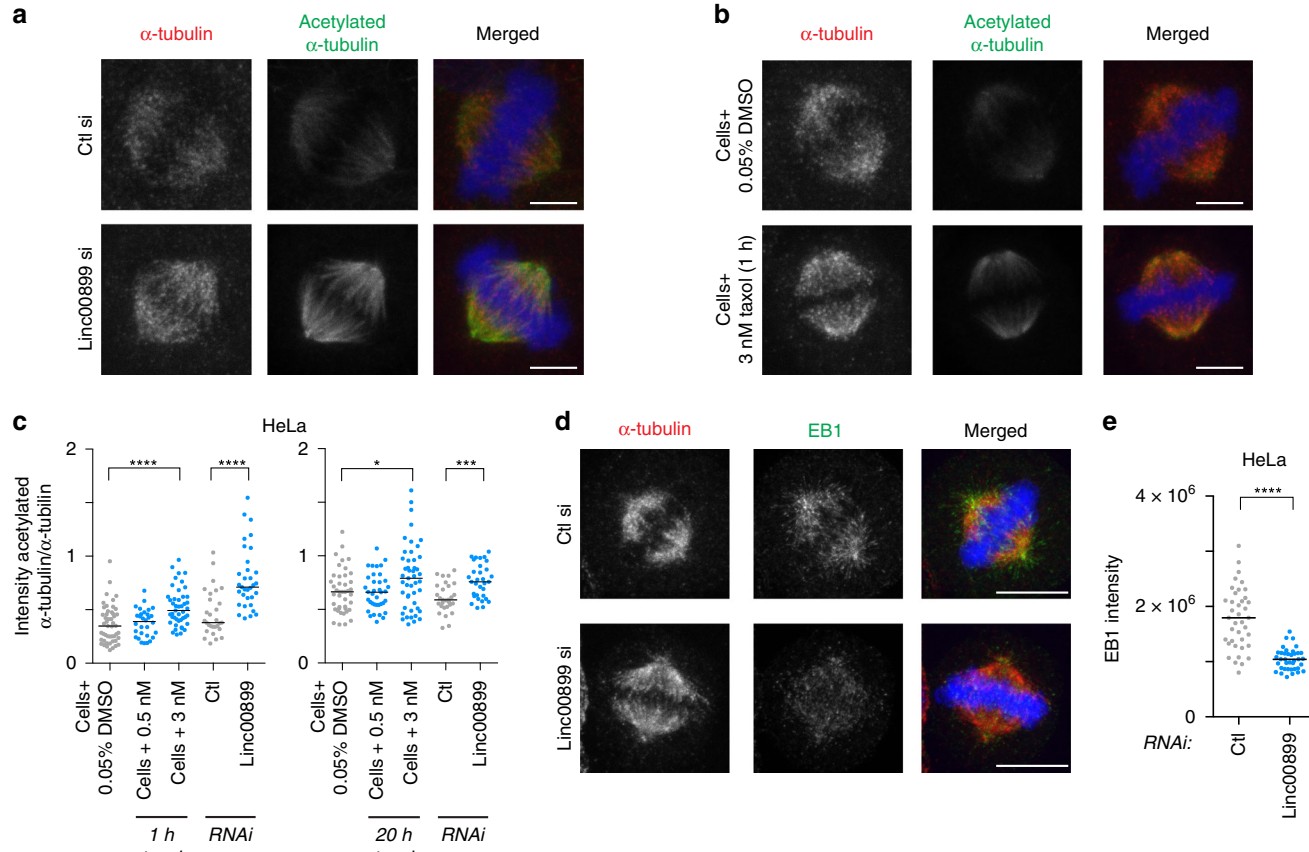

**Fig. 5 Depletion of *linc00899* leads to long-lived mitotic microtubules. a** Acetylated α-tubulin levels in HeLa cells after RNAi-mediated *linc00899* depletion or treatment with a negative control siRNA (Ctl, from Ambion), based on immunofluorescence after staining with antibodies against α-tubulin (microtubules, red) and acetylated α-tubulin (long-lived microtubules, green). Representative images correspond to maximum intensity projections of confocal micrographs. DNA (blue) was stained with Hoechst. **b** Acetylated α-tubulin levels after treatment of HeLa cells with 0.05% DMSO or 3 nM taxol (1 h) based on immunofluorescence as in **a**. Scale bar, 5 µm for **a**, **b**. **c** Quantification of acetylated α-tubulin intensity over the total level of α-tubulin from maximum intensity projections obtained in **a**, **b**. Numbers of cells analysed is $n = 49$ for cells treated with 0.05% DMSO (1 h), $n = 30$ for cells treated with 0.5 nM taxol (1 h), $n = 45$ for cells treated with 3 nM taxol (1 h), $n = 29$ for cells treated with Ctl si and $n = 33$ for *linc00899* RNAi. We also quantified acetylated α-tubulin after 20-h taxol treatment. Numbers of cells analysed is $n = 42$ for cells treated with 0.05% DMSO (20 h), $n = 37$ for cells treated with 0.5 nM taxol (20 h), $n = 47$ for cells treated with 3 nM taxol (20 h), $n = 29$ for cells treated with Ctl si and $n = 32$ for *linc00899* RNAi. Cells with multipolar spindles were excluded from the analysis. Swarm plots represent values from single mitotic cells with the median denoted by the horizontal line. *$P < 0.05$, ***$P < 0.001$ and ****$P < 0.0001$ by Mann–Whitney test. **d** EB1 signal in HeLa cells after RNAi-mediated *linc00899* depletion or treatment with Ctl si based on immunofluorescence after staining with antibodies against α-tubulin (in red) and EB1 (in green). Images correspond to maximum intensity projections of confocal micrographs. DNA is shown in blue. Scale bar, 10 µm. **e** Quantification of EB1 intensity from maximum intensity projections obtained in **d**. Numbers of cells analysed is $n = 39$ for Ctl si and $n = 38$ for *linc00899* RNAi. Swarm plots represent values from single mitotic cells, with the median represented by the horizontal line. ****$P < 0.0001$ by Mann–Whitney test. Source data are provided as a Source Data file.

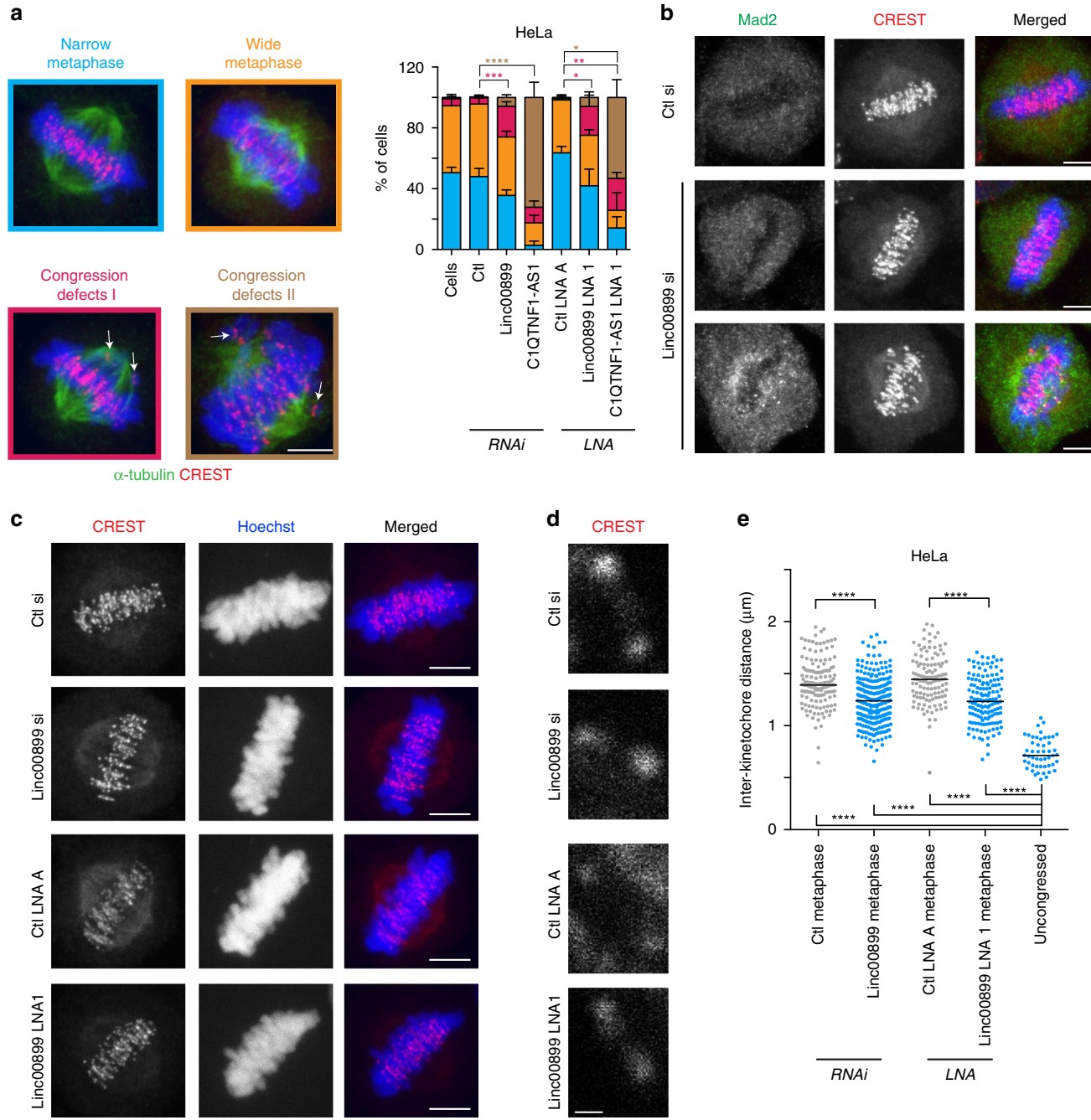

**Fig. 6 Depletion of *linc00899* causes chromosome congression defects. a** Characterisation of mitotic phenotypes in HeLa cells after RNAi- and LNA-mediated depletion of *linc00899* and *C1QTNF1-AS1*. Percentage of mitotic cells with (i) narrow metaphase plate (blue), (ii) wide metaphase plate (orange), (iii) congression defect I (dark pink), and (iv) congression defect II (brown) using α-tubulin (microtubules, in green) and CREST (kinetochore marker, in red). Data are shown as mean ± S.E.M from at least three biological replicates, *P < 0.05, **P < 0.01, ***P < 0.001 and ****P < 0.0001 by two-tailed Student's *t* test. Scale bar, 5 μm. **b** Mad2 is present on kinetochores of unaligned chromosomes in *linc00899*-depleted mitotic cells. Representative images of metaphase cells after RNAi-mediated depletion of *linc00899*. Cells were stained for Mad2 (SAC protein, in green) and with CREST (red). In cells treated with negative control siRNA (Ctl), Mad2-positive kinetochores were detectable in 20% of cells with narrow metaphase plates, whereas this figure was 12% in *linc00899*-depleted cells. In all, 89% of *linc00899*-depleted cells with congression defects were positive for Mad2. DNA was stained with Hoechst and is shown in blue in the merged images. Scale bar, 5 μm. **c** Representative images representing maximum intensity projections of interkinetochore (inter-KT) distance in HeLa cells after RNAi- and LNA-mediated depletion of *linc00899*. Cells were stained with CREST (red) and Hoechst (blue). Scale bar, 5 μm. **d** Insets show examples for individual KT pairs from single focal planes based on CREST staining. Scale bar, 0.5 μm. **e** Quantification of inter-KT distance in cells after *linc00899* depletion as in **c**. A total of 152, 270, 121, 140 and 50 kinetochore pairs (from left to right; at least 7 pairs per cell) were measured for each condition. Numbers of metaphase cells analysed is n = 18 for Ctl si, n = 26 for *linc00899* RNAi, n = 14 for Ctl LNA A, n = 13 for *linc00899* LNA1, and n = 15 for cells with uncongressed chromosomes from *linc00899*-depleted cells. For each condition, we show the median with interquartile range. ****P < 0.0001 one-way ANOVA test. Source data are provided as a Source Data file.

(Fig. 6e). Thus depletion of *linc00899* leads to an overall decrease in tension across sister kinetochores.

In summary, we have demonstrated that *linc00899* depletion suppresses microtubule dynamics, causes a chromosome congression defect and reduces tension on the metaphase plate. These defects are expected to preclude timely inactivation of the SAC, thus delaying anaphase onset.

**Linc00899 regulates *TPPP* and mitosis in multiple cell lines.** We next asked whether *linc00899*-mediated regulation of *TPPP* occurs in cell lines other than HeLa. RNAi-mediated depletion of *linc00899* resulted in upregulation of *TPPP* in three normal diploid cell lines (hTERT-RPE1, retinal pigment epithelial cells; MCF10A, untransformed breast epithelial cells; HUVEC, primary human umbilical vein endothelial cells) where mitosis is ~25 min (Fig. 7a, c, e). Elevated *TPPP* levels were accompanied with a mitotic delay from ~8 to 16 min upon *linc00899* knockdown (Fig. 7b, d, f). This delay was smaller compared to a mitotic delay in HeLa cells and is most likely due to the presence of at least 82 chromosomes in HeLa cells that need to be aligned at the metaphase plate, compared to 46 chromosomes present in normal diploid cells. Indeed, the mitotic timing in HeLa cells is at least ~15 min longer than in RPE1, MCF10A and HUVEC cells. Thus *linc00899* regulates mitotic progression by controlling *TPPP* levels in multiple cell lines.

In addition to its mitotic functions, *TPPP* is crucial for microtubule organisation in the brain. Although TPPP is present in multiple tissues, expression of *TPPP* is highest in the brain in both mouse[48] and human (Supplementary Fig. 12a). In GTEx brain samples, high *TPPP* expression is accompanied by low levels of *linc00899*, and in multiple sclerosis patients *TPPP* and *linc00899* expression were negatively correlated (Supplementary Fig. 12b, c). This suggests that the regulatory relationship observed in our study may be physiologically relevant in human brain tissue and neuropathological diseases.

**Gain-of-function and rescue studies of linc00899.** *Linc00089* could regulate *TPPP* expression in *cis* (locally) or in *trans* (distally), as shown for multiple lncRNAs[3,5,49]. Despite *linc00089* and *TPPP* being located on different chromosomes, a *cis*-acting mechanism whereby lncRNA interacts with its target site by being tethered to its site of synthesis cannot be excluded. To elucidate the mode of action by which *linc00899* regulates *TPPP*, we compared the effects of ectopic and endogenous overexpression of *linc00899* on *TPPP* levels. Despite an increase in *linc00899* expression after its ectopic overexpression using the expression plasmid encoding *linc00899* cDNA, no changes in *TPPP* levels were observed in HeLa and RPE1 cells (Fig. 8a). These results suggest that *linc00899* is unlikely to function in *trans* to regulate *TPPP*. Indeed, rescue experiments using expression plasmid after RNAi- or LNA-mediated depletion of *linc00899* failed to reduce *TPPP* levels (Fig. 8b). Since *linc00899* and *TPPP* are present on different chromosomes, we tested the effect of *linc00899* activation in the context of its normal genomic environment. For that purpose, we employed the CRISPR activation (CRISPRa) system[50], which uses catalytically inactive dCas9 fused to transcriptional activator VP64, and gRNAs targeting different regions of the *linc00899* promoter. We observed twofold overexpression of *linc00899* from its endogenous locus, which led to downregulation of *TPPP* in the normal diploid RPE1 but not in HeLa cells (Fig. 8c). The lack of *TPPP* downregulation in HeLa cells may stem from the presence of multiple *linc00899* and *TPPP* alleles; for instance, CRISPRa may not induce overexpression of all (four or more) *linc00899* loci. In summary, our data suggest

that *linc00899* needs to be expressed from its own locus in order to repress *TPPP* expression.

**Linc0889 binds and regulates transcription of *TPPP*.** As shown in Fig. 2, *linc00899* is a nuclear- and chromatin-enriched lncRNA, raising the possibility that it could directly regulate transcription of *TPPP*. To test this, we performed cleavage under targets and release using nuclease (CUT&RUN)[51] that allows genome-wide profiling of active and repressive histone modifications as well as transcription factors from low cell numbers. We observed an increase in trimethylation of lysine 4 on histone H3 (H3K4me3), a mark of active transcription, at the promoter of *TPPP* after RNAi- and LNA-mediated depletion of *linc00899* (Fig. 9a). Thus elevated levels of *TPPP* mRNA in *linc00899*-depleted cells is likely to arise from increased transcription at the *TPPP* locus. No significant changes were seen in other active and repressive histone modifications at the *TPPP* locus (Supplementary Fig. 13).

Based on these findings, we further hypothesised that *linc00899* binds to regulatory regions of the *TPPP* locus. Therefore, we performed capture hybridisation analysis of RNA targets with sequencing (CHART-seq), a method used to identify lncRNA binding sites on chromatin[52–55]. We first mapped antisense oligonucleotides whose binding is accessible to *linc00899* transcript (Supplementary Fig. 14a) and using the antisense cocktail found approximately tenfold enrichment of *linc00899* transcript compared to the control oligonucleotides (Fig. 9b). No enrichment upon pulldown was detected with the negative control transcript *5.8S*. After sequencing the enriched genomic DNA using the CHART-seq protocol[56], we identified a prominent *linc00899*-binding site in the intron of *TPPP* (Fig. 9c; Supplementary Fig. 14b-d); however, we were not able to identify regions with sequence complementarity to the *linc00899* transcript (Supplementary Fig. 15). Thus transcriptional upregulation of *TPPP* is likely to be mediated by *linc00899* binding to the *TPPP* locus through protein interactions. This could occur through a 3D proximity-guided localisation mechanism, which allows low-abundant lncRNAs, such as *linc00899*, to identify its target genes even on different chromosomes[57]. Such proximity-guided search has been observed for *Firre* and *CISTR-ACT* lncRNAs[58,59] and could explain how *linc00899*, which is encoded on chromosome 22, may bind and regulate *TPPP* encoded on chromosome 5.

To corroborate that *linc00899* act as a transcriptional repressor of *TPPP*, we performed co-RNA FISH using intronic probes against the premature *TPPP* and the mature *linc00899* transcripts (Fig. 9d). *TPPP* has eight alleles in the HeLa cell genome (https://cansar.icr.ac.uk/cansar/cell-lines/HELA/copy_number_variation), yet premature *TPPP* transcripts were detected only in one third of the cells and presented mostly as a single focus. Approximately 3% of cells showed colocalisation between mature *linc00899* and premature *TPPP* transcript. Given that mature *linc00899* transcripts can be detected in most cells, this low level of colocalisation is consistent with effective *linc00899*-mediated suppression of *TPPP* transcription.

## Discussion
Previous studies have attributed the regulation of the cell cycle primarily to multi-protein networks. Here we performed a high-content imaging screen to identify lncRNAs with functions in cell division. Development of in-house image analysis pipelines coupled with targeted validation of lncRNA-induced phenotypes allowed us to quantify the impact of lncRNA depletion on cell division. Among other lncRNAs, this study identified *linc00899* and *C1QTNF1-AS1* as lncRNAs involved in the control of mitotic progression.

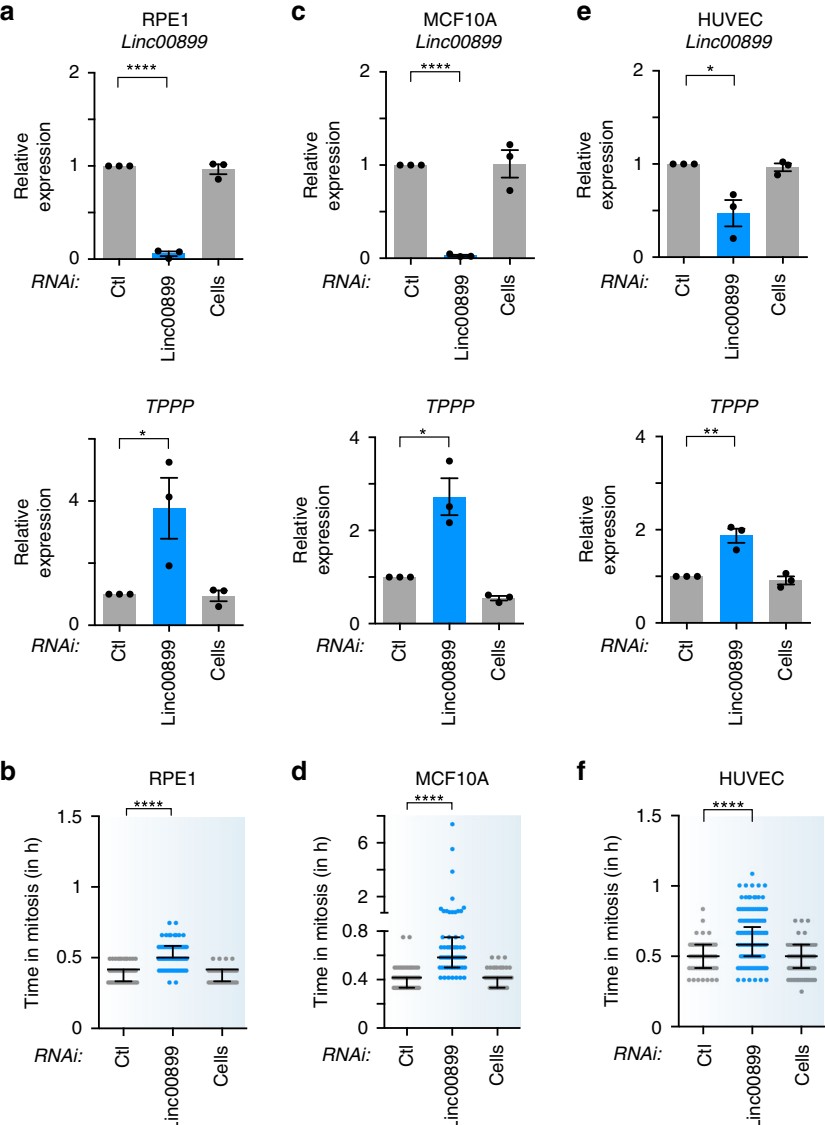

**Fig. 7 *Linc00899* regulates *TPPP* and mitotic progression in different cell lines. a** Expression of *linc00899* and *TPPP* after RNAi-mediated depletion of *linc00899* in normal retina cell line (hTert-RPE1) based on qPCR. Results are also shown for negative control siRNAs (Ctl, from Ambion) and cells treated with transfection reagent alone (Cells). $n = 3$ biological replicates, *$P < 0.05$ and ****$P < 0.0001$ by two-tailed Student's $t$ test. **b** Quantification of mitotic duration from time-lapse microscopy imaging after depletion of *linc00899* in RPE1 cells. Number of cells analysed is $n = 135$ for Ctl si, $n = 98$ for Cells and $n = 103$ for *linc00899* RNAi. Data are shown as median with interquartile range from two biological replicates. ****$P < 0.0001$ by Mann–Whitney test. **c** Expression of *linc00899* and *TPPP* after RNAi-mediated depletion of *linc00899* in non-tumorigenic epithelial breast cells (MCF10A). Controls are as described in **a**. $n = 3$ biological replicates, *$P < 0.05$ and ****$P<0.0001$ by two-tailed Student's $t$ test. **d** Quantification of mitosis duration from time-lapse microscopy imaging after depletion of *linc00899* in MCF10A cells, as described in **c**. Number of cells analysed is $n = 143$ for Ctl si, $n = 142$ for Cells and $n = 63$ for *linc00899* siRNAs. Data are shown as median with interquartile range from two biological replicates. ****$P < 0.0001$ by Mann–Whitney test. **e** Expression of *linc00899* and *TPPP* after RNAi-mediated depletion of *linc00899* in normal primary umbilical vein endothelial cells (HUVEC). Controls are as described in **a**. $n = 3$–4 biological replicates, *$P < 0.05$ and **$P < 0.01$ by two-tailed Student's $t$ test. **f** Quantification of mitotic duration from time-lapse microscopy imaging after depletion of *linc00899* in HUVEC cells, as described in **e**. Number of cells analysed is $n = 58$ for Ctl si, $n = 129$ for Cells and $n = 181$ for *linc00899* RNAi. Data are shown as median with interquartile range from three biological replicates. ****$P < 0.0001$ by Mann–Whitney test. Depletion of *linc00899* in HUVEC, RPE1 and MCF10A leads to an ~8-, 12- and 16-min delay compared to cells treated with Ctl si, respectively. Data are shown as mean ± S.E.M for **a**, **c**, **e**. Mitotic duration in **b**, **d**, **f** was defined from NEBD ($t = 0$ min) to anaphase using bright-field microscopy. Source data are provided as a Source Data file.

Our results revealed that *linc00899* controls mitotic progression by regulating *TPPP*, a protein that binds and stabilises the microtubule network at all stages of the cell cycle[35–38,48,60]. *TPPP* also binds to and inhibits histone deacetylase 6, an enzyme responsible for tubulin deacetylation. This binding results in increased tubulin acetylation[36], a phenotype also observed upon *linc00899* depletion. Fine-tuning TPPP protein levels seems

particularly important for mitosis. Indeed, TPPP overexpression in human cells suppresses microtubule growth velocity and normal microtubule dynamics, thereby impeding timely spindle assembly and cell division[36]. Previous studies have shown that *TPPP* levels are subject to regulation by microRNAs[61] and protein kinases[48,60,62]; our study now reveals lncRNA-mediated transcriptional control as an additional regulatory layer.

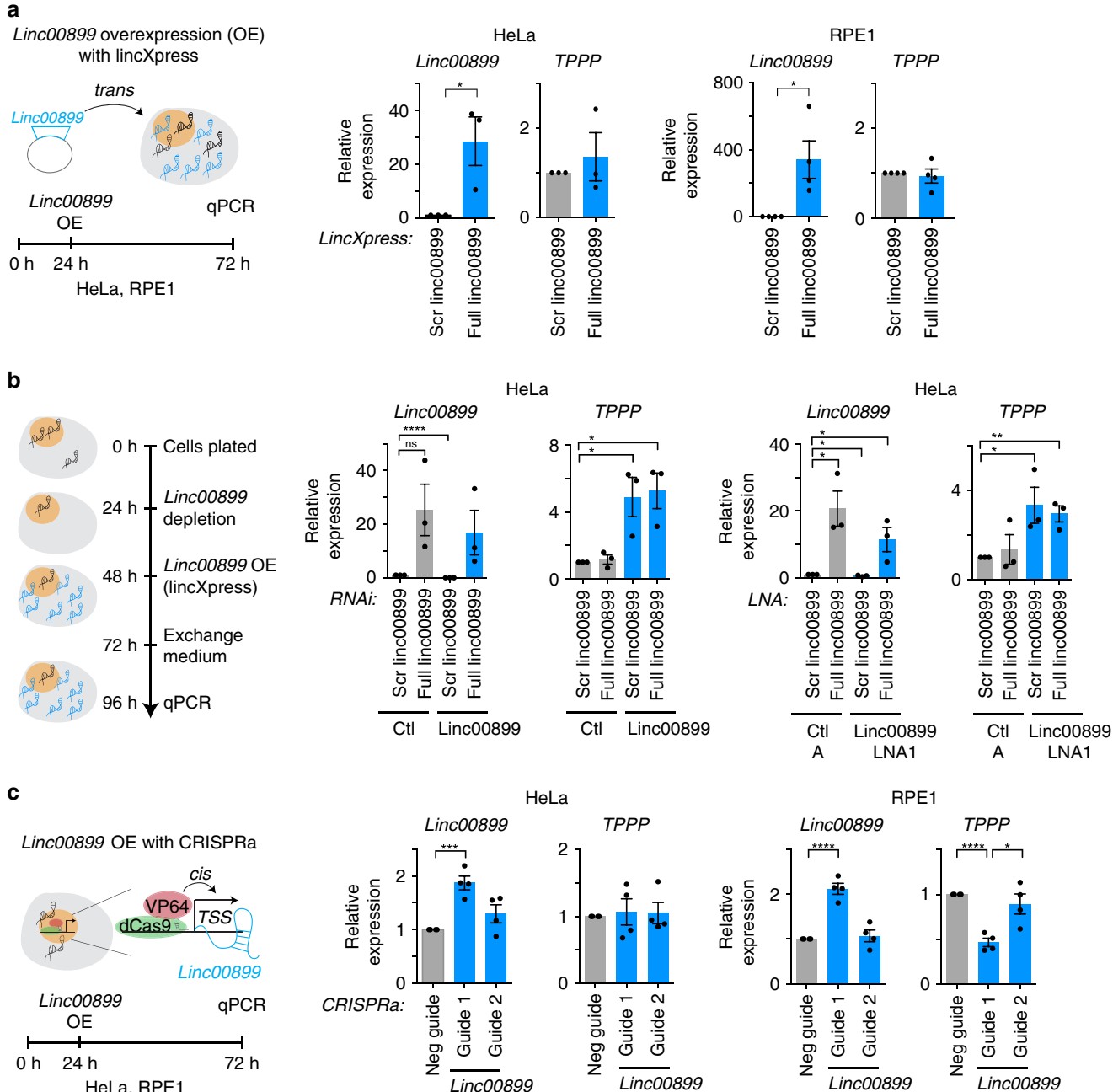

**Fig. 8 Gain-of-function and rescue studies of *linc00899* and its effect on *TPPP* expression. a** Schematic diagram of ectopic overexpression of *linc00899*. *Linc00899* and *TPPP* expression were analysed by qPCR after lentiviral overexpression using lincXpress vector encoding *linc00899* cDNA in HeLa (left) and RPE1 cells (right). The expression was normalised to the scrambled *linc00899* vector (negative control). *n* = 3–4 biological replicates, *\*P* < 0.05 by two-tailed Student's *t* test. **b** Rescue of *linc00899* function after RNAi- or LNA-mediated depletion of *linc00899* in HeLa cells through ectopic overexpression of *linc00899*. *n* = 3 biological replicates, *\*P* < 0.05, *\*\*P* < 0.01 and *\*\*\*\*P* < 0.0001 by two-tailed Student's *t* test. **c** Schematic diagram of endogenous overexpression of *linc00899* using CRISPRa. *Linc00899* and *TPPP* expression were analysed by qPCR after transduction of dCas9-VP64 and gRNAs targeting different regions of the *linc00899* promoter in HeLa (left) and RPE1 cells (right). The expression was normalised to the negative guide (negative control). *n* = 4 biological replicates, *\*P* < 0.05, *\*\*\*P* < 0.001 and *\*\*\*\*P* < 0.0001 by two-tailed Student's *t* test. Data are shown as mean ± S.E.M for **a**–**c**. Source data are provided as a Source Data file.

Our mechanistic studies indicate that *linc00899* regulates *TPPP* transcription. In particular, this notion is supported by our findings that (i) *linc00899* is a nuclear and chromatin-enriched lncRNA, (ii) *linc00899* needs to be expressed from its own locus in order to repress *TPPP* expression, (iii) H3K4me3 levels increase at the *TPPP* promoter in *linc00899*-depleted cells, and (iv) *linc00899* binds at the *TPPP* genomic locus. It is possible that *linc00899* contributes to the repressive chromatin landscape at the

*TPPP* locus by altering local chromatin accessibility as observed with other lncRNAs[63]. The mechanism whereby *linc00899* binds to the *TPPP* genomic locus and represses its transcription remains to be fully defined. Given the limited colocalisation between *linc00899* and the premature *TPPP* transcript, it is more likely that *linc00899* uses "proximity-guided search" where transcription site of *linc00899* resides in close spatial proximity of *TPPP* in the nucleus. This would allow *linc00899* to act

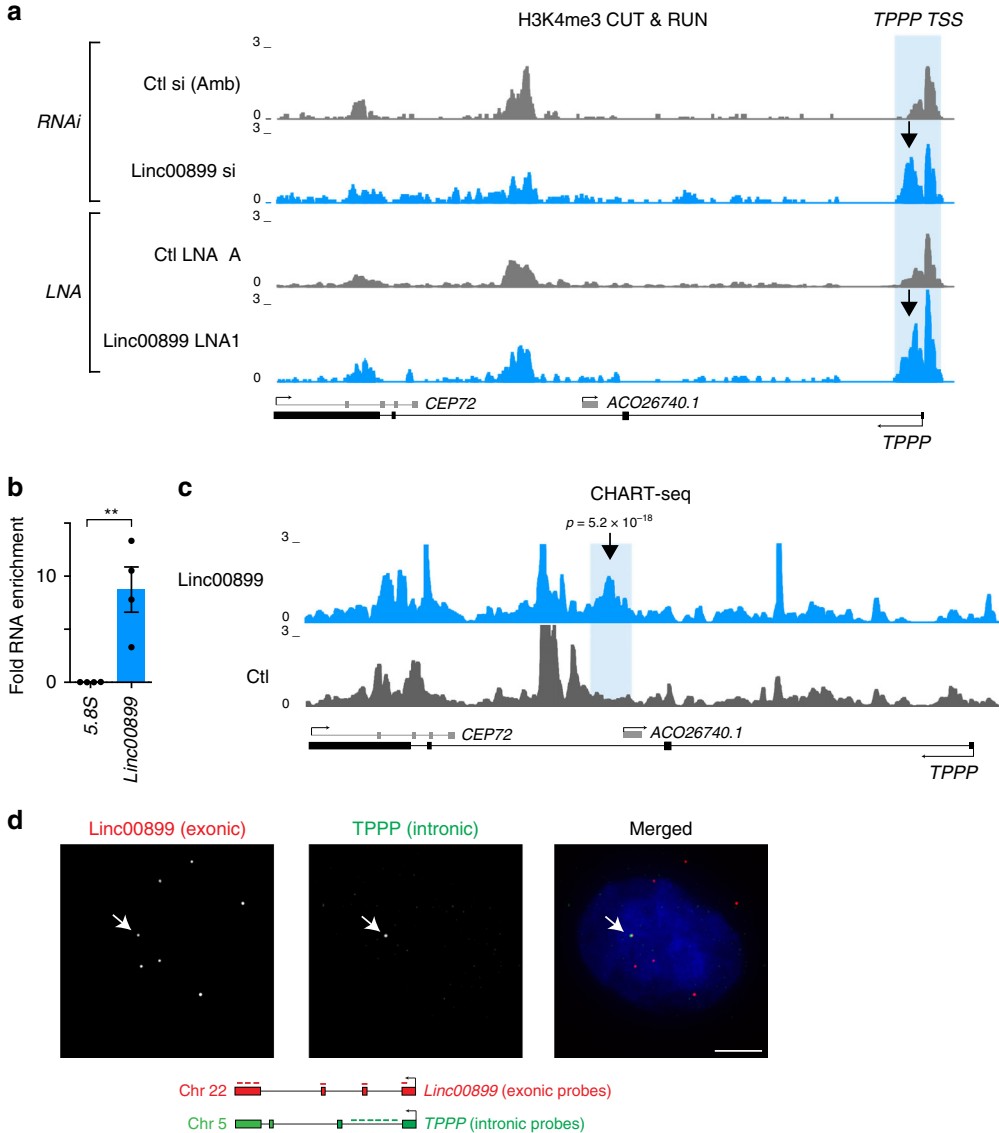

**Fig. 9 *Linc00899* binds to and represses transcription of *TPPP*. a** CUT&RUN profiling of H3K4me3 enrichment after depletion of *linc00899* with RNAi and LNAs at the *TPPP* locus in HeLa cells. An increase in H3K4me3 (arrow) was observed at the *TPPP* promoter upon depletion of *linc00899* with both LOF methods. Tracks were constructed from averages of two biological replicates for each condition. TSS, transcriptional start site. **b** Enrichment of *linc00899* using a cocktail of four oligonucleotides complementary to accessible regions of *linc00899* (as determined by RNase H mapping, Supplementary Fig. 14a), compared to a mixture of two control oligonucleotides containing parts of the *linc00899* sense sequence, as measured by qPCR. The *5.8S* was used as a negative control region. Data are shown mean ± S.E.M. $n = 4$ biological replicates. \*\**P* < 0.01 by two-tailed Student *t* test. **c** Profiling of *linc00899* binding by CHART-seq after RNase H elution at the *TPPP* locus in HeLa cells. The region with the most significant increase in *linc00899* binding compared to the sense control was present in the second intron of *TPPP* (arrow). All binding sites were detected using an empirical FDR threshold of 30%. **d** Validation of *linc00899* colocalisation with the *TPPP* locus by co-RNA FISH in HeLa cells. *Linc00899* (exonic probes against the mature transcript) colocalised to the genomic region of *TPPP* (intronic probes against the premature transcript) in ~3% of the cells ($n = 376$). The nucleus was stained with DAPI (blue). Scale bar, 5 μm. Source data are provided as a Source Data file.

immediately upon its transcription and suppress *TPPP* expression possibly via interactions with protein complexes, such as chromatin regulators[57]. It is also possible that the *linc00899*-mediated regulation of *TPPP* transcription depends on *linc00899* release from the chromatin, with *linc00899* target specificity being guided by the pre-established chromosomal proximity, as shown for lncRNA *A-ROD*[64]. As we did not observe a strong sequence complementarity between the *linc00899* transcript and the *TPPP* DNA sequence, we excluded the possibility of direct binding of *linc00899* to *TPPP* locus. Instead, our data suggest that the *linc00899* function could be mediated through *linc00899*–protein interactions. Further studies will be required to determine the

in vivo *linc00899*–protein interactome and the relevance of these interactions in *TPPP* transcriptional regulation and cell division.

*TPPP* is crucial for microtubule organisation in the brain and for local microtubule nucleation and myelin sheath elongation[65]. In *Drosophila*, *TPPP* mutants (also known as *Ringmaker*) exhibit defects in axonal extension[66]. In mammals, it is primarily expressed in oligodendrocytes where it stabilises microtubule networks, and its depletion in progenitors inhibits oligodendrocyte differentiation[61]. Indeed, *TPPP*-deficient mice display convulsive seizures and motor coordination deficits[65] due to hypomyelination in multiple brain regions, consistent with a defect in myelinating oligodendrocytes[67].

In humans, an inverse correlation between *TPPP* and *linc00899* across all tissues is consistent with a regulatory relationship, with the highest expression of *TPPP* (and lowest expression of *linc00899*) being observed in the brain. This suggests that *linc00899*-dependent suppression of *TPPP* could be used to fine-tune *TPPP* expression and hence microtubule behaviour in a developmental stage- and tissue-specific manner. Intriguingly, altered TPPP protein levels have been observed in a number of neurodegenerative disorders, including multiple sclerosis and Parkinson's disease[68,69].

In this study, we have comprehensively explored the activity of thousands of lncRNAs in cell cycle regulation and identified an assortment of lncRNAs that are involved in controlling mitotic progression, chromosome segregation and cytokinesis. While our analysis encompassed several cellular features, the imaging data from our screen allows us to extract phenotypes at any stage of cell division upon lncRNA depletion. As interest in the regulatory functions of lncRNAs increases, we anticipate that our data will serve as a powerful resource for prioritising lncRNA candidates for further studies in the RNA and cell cycle fields.

## Methods

**Cell lines and reagents**. HeLa and 293FT cells were maintained in Dulbecco's modified Eagle's medium (DMEM; Gibco, 41966-029) supplemented with 10% foetal bovine serum (FBS; Thermo Fisher Scientific, 10500064). HeLa cell line was chosen for this study as majority of phenotypic screens, which identified protein-coding genes in cell division, have been performed in HeLa cells (e.g. Mito-Check[18]). MCF10A (human breast epithelial cell line) were cultured in Mammary Epithelial Cell Growth Medium (Lonza, CC-3151) with supplements and growth factors (Lonza, CC-4136). HUVECs (CC2517, LOT 0000482213) were maintained in Endothelial Cell Growth Medium Lonza (Lonza, CC-3121) with supplements and growth factors (Lonza, CC-4133). RPE1 cells were maintained in Dulbecco's modified Eagle's medium F12 Nutrient Mixture (Gibco, 31331-028) supplemented with 10% FBS (Thermo Fisher Scientific, 10500064, LOT 2025814K). All cell lines were obtained from the American Type Culture Collection (ATCC) and were cultured at 37 °C with 5% CO$_2$. HeLa Kyoto (EGFP-α-tubulin/ H2B-mCherry) cells were obtained from ATCC/Jan Ellenberg (EMBL Heidelberg)[18] and cultured in DMEM with 10% FBS. All cell lines were verified by short tandem repeat profiling and tested negative for mycoplasma contamination. Paclitaxel (taxol) used in cell culture experiments was dissolved in dimethyl sulfoxide (DMSO; T7402, Sigma) and used at final concentrations of 0.5 or 3 nM. The cells were treated for 1 or 20 h. Final concentration of DMSO in media was 0.05%.

**High-content imaging screen: Lincode siRNA library**. The Lincode siRNA Library (GE Dharmacon, G-301000) is a collection of siRNA reagents targeting 2231 human lncRNAs (1860 unique human lncRNA genes and 371 lncRNA transcripts associated with protein-coding genes). The design of this library is based on RefSeq version 54 and the siRNAs are arrayed as SMARTpools. The library was purchased at 0.1 nmol in a 96-well format. The library was diluted to a 5 μM stock with 1× siRNA buffer (GE Dharmacon, B-002000-UB-100) and arrayed onto black 384-well PerkinElmer Cell Carrier plates (Perkin Elmer, 6007550) using the Echo Liquid Handler (Labcyte). The black 384-well PerkinElmer Cell Carrier plates were prepared in advance and stored at −80 °C. The final siRNA concentration per well was 20 nM. An siRNA targeting exon 1 of lncRNA *GNG12-AS1* (Silencer select, Life Technologies, S59962)[70] and SMARTpool siRNAs targeting protein-coding gene *CKAP5/Ch-TOG* (GE Dharmacon, L-006847-00) were also included on each plate. *CKAP5/Ch-TOG* was used as a positive control as its depletion leads to mitotic delay and increased mitotic index[24]. *ECT2* SMARTpool siRNAs (GE Dharmacon, L-006450-00-0005) were used as a positive control in the third validation screen as its depletion results in multinucleated cells[26].

**High-content imaging screen: reverse transfection**. To redissolve the siRNA in the black 384-well PerkinElmer Cell Carrier plates (10 plates in total), 5 μl of OptiMEM medium (Thermo Fisher Scientific, 31985047) was added. The plates were centrifuged (1 min, 900 × *g*) and incubated at room temperature (RT) for 5 min. Lipofectamine RNAimax (Thermo Fisher Scientific, 13778150) was added in OptiMEM medium to a final concentration of 8 μl Lipofectamine to 1 ml OptiMEM medium and incubated at RT for 5 min. Five μl of OptiMEM/Lipofectamine mix was then added to the plates. Plates were centrifuged and incubated at RT for 20 min. In the meantime, HeLa cells were trypsinised and counted using an automated cell counter (Countess, Thermo Fisher Scientific). Cells were centrifuged at 1000 × *g* for 4 min, the medium was removed and the cells were resuspended in OptiMEM medium to a final concentration of 2000 cells/well. Ten μl of cell suspension was added to the plates, and plates were centrifuged and incubated at 37 °C for 4 h, before adding 10 μl of DMEM+30% FBS+3% P/S

(penicillin/streptomycin, P/S) (final concentration 10% FBS, 1% P/S). Plates were centrifuged and incubated at 37 °C for 48 h before fixation. A Multidrop Combi Reagent dispenser (Thermo Fisher Scientific) was used throughout the transfection protocol to ensure even liquid addition.

**High-content imaging screen: fixation and immunostaining**. For screen A, the plates were fixed by adding an equal volume of pre-warmed (37 °C) 8% formaldehyde (Thermo Fischer Scientific, 28908)/phosphate-buffered saline (PBS) solution to the wells and incubated at 37 °C for 10 min. The cells were permeabilised with pre-warmed PBS/0.2% Triton X-100 (Acros Organics, 327371000) for 15 min at RT. The cells were then blocked in 1% bovine serum albumin (BSA)/PBS for 1 h at RT. To perform the immunostaining, the cells were incubated with primary antibodies against α-tubulin (Dm1α, Sigma, TUB9026, dilution 1:1000), CEP215/CDK5RAP2[71] (dilution 1:500) and Alexa-Fluor® 568 Phalloidin (Thermo Fisher Scientific, A12380, dilution 1:500) for 2 h at RT. The cells were washed three times in 1× PBS and incubated for 1 h at RT with secondary antibodies Alexa Fluor® 488 (Thermo Fisher Scientific, A21206, dilution 1:1000) and Alexa Fluor® 647 (Thermo Fisher Scientific, A31571, dilution 1:1000). After three washes in 1× PBS, the cells were incubated with 1 μg/ml Hoechst (Sigma, H 33258, diluted in PBS) for 15 min at RT before a final wash in 1× PBS and imaging.

For screen B, the same fixation protocol as described for screen A was used. For permeabilisation, PBS/0.05% sodium dodecyl sulfate was used for 20 min at RT. Blocking was performed as described above. For the immunostaining, cells were incubated with primary antibodies against γ-tubulin (Sigma, GTU88, dilution 1:1000) and phospho-histone H3 serine 10 (PHH3, Millipore, 06-570, dilution 1:2000) for 2 h at RT and washed three times with 1× PBS before incubation with secondary antibodies (Thermo Fisher Scientific, A-31571 and A-21206, both at dilution 1:1000). After three 1× PBS washes, the cells were stained with α-tubulin (Serotec, MCA78G, dilution 1:500) and incubated with secondary antibody (Thermo Fisher Scientific, A-21434, dilution 1:1000). All primary and secondary antibodies were diluted in 1% BSA/PBS. The fixation and staining for both replicates was carried out at the Institute for Cancer Research (ICR, London) using the PerkinElmer Cell:Explorer system coupled to automated liquid handling equipment. Solutions were dispensed using a Multidrop Combi Reagent dispenser (Thermo Fisher Scientific) and aspirated/washed using a Biotek washer with 96 pins. All plates were imaged using the PerkinElmer Opera high-content confocal screening platform with spinning disc. Thirty fields of view per well were captured using a ×20 air objective, numerical aperture (NA) 0.45.

**High-content imaging screen: image analysis**. All image analysis was performed using custom workflows created with the Columbus software (PerkinElmer). Several output parameters were evaluated from high-content images: mitotic index (number of cells in mitosis), multinucleation index (number of multinucleated cells), number of viable cells, number of chromosome segregation errors (chromatin bridges and lagging chromatids) and number of cells with cytokinetic bridges (Supplementary Fig. 1). These are defined in more detail below.

**Mitotic and multinucleation index**. Nuclei were first segmented using Hoechst staining (which defines the total cell number). The false positives (e.g. dead cells) were discarded based on the nucleus area, α-tubulin and γ-tubulin/ CEP215 staining intensity. Multinucleated non-dividing cells were retained as a separate subpopulation using a two-step detection process: binucleated cells were isolated using size, aspect ratio and roundness parameters of close nucleus pairs. Other multinucleated cells were then identified among remaining cells for which α- or γ-tubulin intensity was low in the perinuclear region. Further identification of mitotic cells/stages was accomplished using filters based on Hoechst for the nucleus shape and size, in combination with high PHH3 (screen B) or high α-tubulin and low CEP215 staining intensity (screen A). Notably, nuclei of cells in anaphase/ telophase stage of mitosis were small, had elongated shape and exhibited low Hoechst integrated intensity (low amount of DNA among mitotic cells). The distance between both nuclei of cells in anaphase/telophase stage was the main criteria to discard two daughter non-mitotic cells (maximum of 0.65 and 2.6 μm, respectively). From all these sub-populations, we calculated mitotic and multi-nucleation index relative to the total number of live cells.

**Chromosome segregation errors**. We started from the previous identified sub-population of cells in anaphase/telophase stage. We filtered cells according to α-tubulin staining intensity between nuclei, as cells in anaphase have lower α-tubulin intensity compared to the cells in telophase. This allowed us to identify only the cells in anaphase. To calculate the number of anaphase cells with chromosome segregation errors, the inter-nuclei space was used as the measuring area to calculate the remaining Hoechst signal. This captures both chromatin bridges and lagging chromatids.

**Number of viable cells**. The total number of viable cells was determined after removal of dead cells and cell debris with anaphase and telophase cells counted as one (despite exhibiting two nuclear segments). The same rule was applied for multinucleated cells.

**Number of cytokinetic bridges**. Cytokinetic bridges were defined as elongated high-intensity objects split into two parts that are positive for α-tubulin staining. We used prior segmentation of the cytoplasm and the Spot Finder feature to identify bridge half parts and sorted them as doublets by calculating the distance between them. We discarded the false positive candidates based on the shape criteria and γ-tubulin staining. The final number of cytokinetic bridges was divided by the total number of viable cells.

To minimise the variation in the cell density between different wells among all ten 384-well plates, we divided the output numbers by the total number of cells per well (dead cells were not included). Multinucleated cells were considered to be single cells during counting. The ratios were then normalised between screen plates by calculating the average value per plate and finally the grand average of all ten plates, giving a reference mean ratio. Per-well ratios were scaled so that the per-plate average was equal to the reference. Z-scores (z) were calculated as follows for each parameter:

$$z = (x - \mu)/\sigma \qquad (1)$$

where x represents the ratio for the feature of interest (e.g. mitotic/multinucleation index), μ represents the reference ratio and σ represents the standard deviation of ratios across wells. All the scripts used for the image analysis are available at https://github.com/MarioniLab/LncScreen2018.

**High-content imaging screen: third-pass validation screen**. The third-pass validation screen was performed in two replicates with four technical replicates using the top 25 candidates from each of the categories (mitotic progression, cytokinesis) using the same antibodies as in screen B. Correlation coefficients between replicate plates in third screen were calculated by Spearman's rank correlation (mitotic index = 0.967888, viability = 0.93249, multinucleation index = 0.995897, cytokinetic bridges = 0.898324).

**Single-molecule RNA FISH**. Cells were grown on coverslips in 12-well plates, briefly washed with 1× PBS and fixed with PBS/3.7% formaldehyde at RT for 10 min. Following fixation, cells were washed twice with PBS. The cells were then permeabilised in 70% ethanol for at least 1 h at 4 °C. Stored cells were briefly rehydrated with Wash Buffer (2× SSC, 10% formamide, Biosearch) (Formamide, Thermo Fisher Scientific, BP227-100) before FISH. The Stellaris FISH Probes (linc00899 and C1QTNF1-AS1 exonic probes Q570) were added to the hybridisation buffer (2× SSC, 10% formamide, 10% dextran sulfate, Biosearch) at a final concentration of 250 nM. Hybridisation was carried out in a humidified chamber at 37 °C overnight. The following day, the cells were washed twice with Wash Buffer (Biosearch) at 37 °C for 30 min each. The second wash contained 4,6-diamidino-2-phenylindole for nuclear staining (5 ng/ml, Sigma, D9542). The cells were then briefly washed with 2× SSC and then mounted in Vectashield (Vector Laboratories, H-1000). Images were captured using a Nikon TE-2000 inverted microscope with the NIS-elements software, a Plan Apochromat ×100 objective and an Andor Neo 5.5 sCMOS camera. We acquired 25 optical slices at 0.3-μm intervals. Images were projected in two dimensions using ImageJ and deconvolved with Huygens Professional.

For validation of CHART-sequencing, Stellaris FISH Probes for intronic region of TPPP (Q670) were combined with linc00899 exonic probe (Q570) at a final concentration of 250 nM per probe set. To score whether TPPP (intronic signal) and linc00899 (exonic signal) colocalize, we only considered cells in which both signals were present. The sequences of RNA FISH probes are presented in Supplementary Data 4.

**Plasmids**. To insert pAS into lncRNAs, pSpCas9(BB)−2A-GFP vector was used (PX458, Addgene, #48138). For CRISPR activation (CRISPRa), pHAGE EF1alpha dCAS-VP64-HA (Addgene, #50918), pU6-sgRNA EF1Alpha-puro-T2A-BFP (Addgene, #60955), second-generation packaging plasmid psPAX2 (Addgene, #12260) and the envelope plasmid pMD2.G (Addgene, #12259) were used. Linc00899 and C1QTNF1-AS1 constructs were synthesised by Labomics. The full sequence of linc00899 was synthesised as CS-LNC233J-T7 (4283 bp, insert was 1610bp based on NR_027036), scrambled linc00899 was CS-LNC236J-T7, full C1QTNF1-AS1 was CS-LNC237JT7 (3638 bp, insert was 970 bp based on NR_040018) and scrambled C1QTNF1-AS1 was CS-LNC238JT7). Sequences of Labomics lncRNA vectors (pUCLOMT) are presented in Supplementary Methods.

**RNA isolation, cDNA synthesis and qPCR**. RNA (1 μg) was extracted with the RNeasy Kit (QIAGEN, 74106) and treated with DNase I following the manufacturer's instructions (QIAGEN, 79254). The QuantiTect Reverse Transcription Kit (QIAGEN, 205313) was used for cDNA synthesis including an additional step to eliminate genomic DNA contamination. qPCR was performed on a QuantStudio 6 Flex (Thermo Fischer Scientific) with Fast SYBR Green Master Mix (Life Technologies). Thermocycling parameters were defined as 95 °C for 20 s followed by 40 cycles of 95 °C for 1 s and 60 °C for 20 s. Two reference genes (GAPDH and RPS18) were used to normalise expression levels using the $2^{-\Delta\Delta CT}$ method. Sequences of qPCR primers are provided in Supplementary Table 1. For linc00899 and C1QTNF1-AS1 primers against exons 2–4 and 1–2, respectively, were used throughout the paper if not indicated otherwise.

**siRNA and LNA depletion experiments**. HeLa cells were transfected with Lipofectamine RNAiMax reagent (Thermo Fischer Scientific) following the manufacturer's instructions. All experiments were done 48 h after transfection. The pool of four siRNA sequences (SMARTpool, Thermo Fischer Scientific) and LNA Gapmers (Exiqon) were used at a final concentration of 50 and 25 nM, respectively. For double knockdown experiments, HeLa cells (10,000 cells/well) were plated on 8-well chamber slides (Ibidi, 80826) for time-lapse microscopy imaging or in 6-well for RNA extraction (Corning, 120,000 cells/well). The cells were transfected the next day with either negative control siRNA (Ctl, from Ambion), a SMARTpool of siRNAs targeting linc00899 or TPPP or siRNAs targeting linc00899 in combination with TPPP. The same concentration of 50 nM was achieved for both single and double knockdown by adding equal amount of control siRNA sequence (Ctl) to the single SMARTpool targeting linc00899 or TPPP separately. siRNA and LNA sequences are listed in Supplementary Tables 2 and 3, respectively.

**Western blot analysis**. Cells were grown in a six-well plate, trypsinised, pelleted and washed twice with 1× PBS. The pellet was lysed in lysis buffer (50 mM Tris-HCl, pH 8, 125 mM NaCl, 1% NP-40, 2 mM EDTA, 1 mM PMSF [Sigma, 93482-50ML-F] and protease inhibitor cocktail [Roche, 000000011836170001]) and incubated on ice for 25 min. The samples were centrifuged for 3 min at 12,000 × g and 4 °C. Supernatant was collected and protein concentration was determined using the Direct Detect® Spectrometer (Merck Millipore). The proteins (30 μg) were denatured, reduced, and separated with Bolt® 4–12% Bis-Tris Plus Gel (Thermo Fisher Scientific, NW04120BOX) in MOPS buffer (Thermo Fisher Scientific, B0001-02). Precision Plus Protein Standards (161-0373, Bio-Rad) was used as a protein standard. The proteins were then transferred to nitrocellulose membrane and blocked with 5% nonfat milk in TBS-T (50 mM Tris, 150 mM NaCl, 0.1% Tween-20) for 1 h at RT. The membranes were incubated with primary antibodies in 5% milk in TBS-T with the following antibodies: TPPP/p25 (NBP2-34031, Novus, dilution 1:1000), β-tubulin (T019, Sigma, dilution 1:2000), p150 (610473, BD Transduction Laboratories, dilution 1:2000), Cyclin B1 (12231S, Cell signalling, dilution 1:1000), RAI14 (NBP1-94075, Novus, dilution 1:300), DNAAF5 (HPA020243, Atlas Antibodies, dilution 1:250), ITGB1BP1 (HPA071538, Atlas Antibodies, dilution 1:250), and phospho Histone H3 serine 10 (06-570, Millipore, dilution 1:1000). After overnight incubation at 4 °C, the membranes were washed with TBS-T and incubated with horseradish peroxidase secondary antibodies (GE Healthcare Life Sciences, 1:2000), and immunobands were detected with a Pierce ECL Western Blotting Substrate (Thermo Fischer Scientific, 32106). Quantification of immunoblots normalised against appropriate loading controls was done using ImageJ. Uncropped scans of the immunoblots are provided in the Source Data file. The list of all primary and secondary antibodies is provided in Supplementary Table 4.

**Time-lapse microscopy imaging**. HeLa (10,000 cells/well), HeLa Kyoto (10,000 cells/well), RPE1 (10,000 cells/well), MCF10A (15,000 cells/well) and HUVEC cells (15,000 cells/well) were cultured in 8-well chamber slides (Ibidi) with 200 μl/well of the corresponding medium. Imaging was performed in their corresponding medium. Time-lapse microscopy imaging was performed for all cell lines 48 h after transfection with RNAi or LNA gapmers. Mitotic duration was measured as the time from nuclear envelope breakdown until anaphase onset, based on visual inspection of the images. Cytokinesis was measured from anaphase onset to abscission completion. Live-cell imaging was performed using a Zeiss Axio Observer Z1 microscope equipped with a PL APO 0.95NA ×40 dry objective (Carl Zeiss Microscopy) fitted with a LED light source (Lumencor) and an Orca Flash 4.0 camera (Hamamatsu). Four positions were placed per well and a z-stack was acquired at each position every 10 min (HeLa, HeLa Kyoto) or 5 min (MCF10A, HUVEC, RPE1) for a total duration of 12 h. To detect chromosome segregation errors (chromatin bridges and/or lagging chromatids), HeLa Kyoto cells were imaged every 4 min with only 2 positions/well. Voxel size was 0.325 μm × 0.325 μm × 2.5 μm. Zen software (Zeiss) was used for data collection and analysis. Throughout the experiment, cells were maintained in a microscope stage incubator at 37 °C in a humidified atmosphere of 5% CO_2.

**Cell cycle synchronisation**. HeLa cells were grown to 50% confluency and then synchronised with thymidine for 16 h (2 mM, Sigma, T1895). The cells were washed three times with 1× PBS and then released into thymidine-free medium for 5 h. The cells were released for the indicated timepoints for RNA collection. Treatment of HeLa cells with monastrol (100 μM for 16 h; Tocris, 1305) coupled with the mitotic shake-off was used to isolate mitotic cells.

**Copy number evaluation**. To calculate the linc00899 and C1QTNF1-AS1 copy number, a standard curve of Ct values was generated by performing qPCR on a dilution series of known concentration of linc00899 or C1QTNF1-AS1 DNA templates (Labomics). cDNA was prepared from RNA (1 μg) extracted from the known number of HeLa cells (500,000 cells). The observed Ct values were fitted on the standard curve and the number of lncRNA molecules per cell was calculated. The final value was multiplied by 2 to account for the fact that cDNA is single stranded and DNA templates used to make the standard curve were double stranded.

**RNA library preparation, sequencing and analysis**. RNA-seq libraries were prepared from HeLa cells using TruSeq Stranded Total RNA Kit with Ribo-Zero Gold (Illumina, RS-122-2303). We generated four biological replicates for RNAi- and LNA-mediated depletion of *linc00899* and *C1QTNF1-AS1*. Indexed libraries were PCR-amplified and sequenced using 125 bp paired-end reads on an Illumina Hiseq 2500 instrument (CRUK CI Genomics Core Facility). Each library was sequenced at a depth of 20–30 million read pairs. Paired-end reads were aligned to the human genome hg38 with subread and the number of read pairs mapped to the exonic regions of each gene was counted for each library by using the feature-Counts[72] function in Rsubread v1.30.0 with Ensembl GRCh38 version 91. Only alignments with mapping quality scores >10 and with the first read pair on the reverse strand were considered during counting. Approximately 80% of read pairs contained one read that was successfully mapped to the reference, and 74% of all read pairs in each library were assigned into exonic regions. Any outlier samples with very low depth (resulting from failed library preparation or sequencing) were removed prior to further analysis.

The DE analysis was performed using the limma package v3.36.0[73]. First, lowly expressed genes with average counts <3 were filtered out. Normalisation was performed using the trimmed mean of *M*-values (TMM) method to remove composition biases. Log-transformed expression values with combined precision/array weights were computed with the voomWithQualityWeights function[74]. The experimental design was parametrised using an additive model with a group factor, where each group was comprised of all samples from one batch/treatment combination, and an experiment factor, representing samples generated on the same day. Robust empirical Bayes shrinkage[75] was performed using the eBayes function to stabilise the variances. Testing for DEGs was performed between pairs of groups using the treat function[76] with a log-fold change threshold of 0.5. Here the null hypothesis was that the absolute $\log_2$-fold change between groups was ≤0.5. All pairwise contrasts involved groups from the same batch to avoid spurious differences due to batch effects. For each contrast, genes with significant differences in expression between groups were detected at an false discovery rate (FDR) of 5%. To identify DEGs that were consistent across LOF methods, an intersection–union test was performed[77] on one-sided *P* values in each direction, followed by an FDR correction. Coverage tracks for each library were generated using Gviz as previously described.

**CHART sequencing and analysis**. CHART enrichment and RNase H (NEB, M0297S) mapping was performed as previously described[52,56]. Briefly, five 150-mm dishes of HeLa cells were used to prepare the CHART extract for each pull-down. In the first sonication step, the samples were sonicated using Covaris S220 (Covaris, 500217) in microTUBES (Covaris, 520045) in a final volume of 130 µl (Programme conditions: 20% duty cycle, 200 bursts/cycle, Intensity 175, 8 min). The extracts were hybridised with a *linc00899* oligonucleotide cocktail (mix of 4 oligos used at final concentration of 25 µM) overnight at RT. MyOne streptavidin beads C1 (Thermo Fischer Scientific, 65001) were used to capture complexes overnight at RT by rotation. Bound material was washed five times and RNase H (NEB) was added for 30 min at RT to elute RNA–chromatin complexes. To increase recovery yield, the remaining beads were saved and the bound material was eluted with Proteinase K (Thermo Fischer Scientific, 25530-049) in 100 µl of XLR buffer for 60 min at 55 °C. The supernatant was then collected and heated for additional 30 min at 65 °C. The RNase H eluate samples were also treated with Proteinase K (Thermo Fischer Scientific, 25530-049) and cross-links were reversed (55 °C for 1 h, followed by 65 °C for 30 min). One fifth of the total sample was used to purify RNA using the miRNeasy Kit (QIAGEN, 217004) and to calculate RNA enrichment, while the rest of the sample was used to purify DNA using Phenol-ChCl₃:isoamyl (Thermo Fischer Scientific, 15593-031) extraction and ethanol precipitation. The final CHART DNA was eluted in 12.5 µl of 1× TE buffer (low EDTA, pH 7.4). Ten µl from this reaction was used for sonication using Covaris LE220 (programme 250 bp/10 s) to obtain average fragment size of 200–300 bp. For this, microTUBE-15 beads strips (520159, Covaris) were used in a total volume of 15 µl. After sonication, the DNA was measured by Nanodrop (input, diluted 1:20) and Qubit High Sensitivity DNA Assay (eluates were undiluted). CHART material (5 ng) was used for library preparation using the ThruPLEX DNA-Seq library preparation protocol (Rubicon Genomics, UK). For inputs, six PCR cycles were performed, while for eluates eight PCR cycles were performed owing to the lower quantity of input DNA. Library fragment size was determined using a 2100 Bioanalyzer (Agilent). Libraries were quantified by qPCR (Kapa Biosystems). Pooled libraries were sequenced on a HiSeq4000 (Illumina) according to the manufacturer's instructions to generate paired-end 150 bp reads (CRUK CI Genomics Core Facility). CHART-seq was performed from five biological replicates using a mix of two sense control oligos for *linc00899* as a negative control for the *linc00899* pulldown. Reads were aligned to the hg38 build of the human genome using subread in paired-end genomic mode. A differential binding (DB) analysis was performed using csaw to identify lncRNA-binding sites in the pulldown compared to the sense control. Filtered windows were obtained as described for the CUT&RUN data analysis. Normalisation factors were computed by binning read pairs into 5-kbp intervals and applying the TMM method without weighting, to account for composition biases from greater enrichment in the antisense pulldown samples. The filtered windows and normalisation factors were then used in a DB analysis with the quasi-likelihood framework in edgeR. This was performed using

an additive design for the generalised linear model fit, containing terms for the batch and the pulldown (five batches in total, sense and antisense pulldowns in each batch). A *P* value was computed for each window by testing whether the pulldown term was equal to zero, i.e. no difference in coverage between the anti-sense pulldown and sense control. The above analysis was repeated using window sizes from 150 to 1000 bp to obtain DB results at varying resolutions. These results were consolidated into a single list of DB regions using consolidateWindows as previously described, yielding a combined *P* value for each region. The combined *P* values across all regions were then used to compute the empirical FDR (eFDR), defined as the ratio of the number of clusters that exhibited increased coverage in the sense control (i.e. false positives) to the number of clusters with increased coverage in the antisense pulldown (i.e. potentially true discoveries). Putative lncRNA-binding sites were defined at an eFDR of 30%. Coverage tracks for each library were generated using Gviz as previously described. The list of CHART probes and CHART primers is provided in Supplementary Tables 5 and 6, respectively.

**CUT&RUN (cleavage under targets and release using nuclease)**. Chromatin profiling was performed according to Skene et al.[51] with minor modifications. HeLa cells were plated in 30-mm dishes (220,000 cells/dish) and transfected the next day with either negative control siRNA, *linc00899* siRNA pool, negative control LNA A or an LNA gapmer targeting *linc00899* (LNA1). Two biological replicates were performed per condition. Cells were washed once with PBS, spun down at $600 \times g$ for 3 min in swinging-bucket rotor and washed twice with 1.5 ml Wash buffer (20 mM HEPES-KOH (pH 7.5), 150 mM NaCl, 0.5 mM Spermidine and 1× cOmplete™ EDTA-free protease inhibitor cocktail (Roche)). During the cell washes, concanavalin A-coated magnetic beads (Bangs Laboratories, BP531) (10 µl per condition) were washed twice in 1.5 ml Binding Buffer (20 mM HEPES-KOH (pH 7.5), 10 mM KCl, 1 mM CaCl, 1 mM MnCl₂) and resuspended in 10 µl Binding Buffer per condition. Cells were then mixed with beads and rotated for 10 min at RT, and samples were split into aliquots according to the number of antibodies profiled per cell type. We used 80–100,000 cells per chromatin mark. Cells were then collected on magnetic beads and resuspended in 50 µl Antibody Buffer (Wash buffer with 0.05% Digitonin and 2 mM EDTA) containing one of the following antibodies in 1:100 dilution: H3K4me3 (Millipore 05-1339 CMA304, Lot 2780484), H3K27me3 (Cell Signaling #9733 S C36B11, Lot 8), H3K36me3 (Active Motif Cat#61101, Lot 32412003), H3K27ac (Abcam ab4729, Lot GR3211741-1) and Goat Anti-Rabbit IgG H&L (Abcam ab97047, Lot GR254157-8). Cells were incubated with antibodies overnight at 4 °C rotating, and then washed once with 1 ml Digitonin buffer (Wash buffer with 0.05% Digitonin). For the mouse anti-H3K4me3 antibody, samples were incubated with 50 µl of a 1:100 dilution in Digitonin buffer of secondary rabbit anti-mouse antibody (Invitrogen, A27033, Lot RG240909) for 10 min at RT and then washed once with 1 ml Digitonin buffer. Samples were then incubated in 50 µl Digitonin buffer containing 700 ng/ml Protein A-MNase fusion protein (kindly provided by Steven Henikoff) for 10 min at RT followed by two washes with 1 ml Digitonin buffer. Cells were then resus-pended in 100 µl Digitonin buffer and cooled down to 4 °C before addition of CaCl₂ to a final concentration of 2 mM. Targeted digestion was performed for 30 min on ice until 100 µl of 2× STOP buffer (340 mM NaCl, 20 mM EDTA, 4 mM EGTA, 0.02% Digitonin, 250 mg RNase A, 250 µg Glycogen, 15 pg/ml yeast spike-in DNA (kindly provided by Steven Henikoff)) was added. Cells were then incubated at 37 °C for 10 min to release cleaved chromatin fragments, spun down for 5 min at $16,000 \times g$ at 4 °C and collected on magnet. Supernatant containing the cleaved chromatin fragments were then transferred and cleaned up using the Zymo Clean & Concentrator Kit. Library preparation[78] was performed using the Thru-PLEX® DNA-Seq Library Preparation Kit (Rubicon Genomics) with a modified library amplification programme: extension and cleavage for 3 min at 72 °C fol-lowed by 2 min at 85 °C, denaturation for 2 min at 98 °C followed by four cycles of 20 s at 98 °C, 20 s at 67 °C, and 40 s at 72 °C for the addition of indexes. Ampli-fication was then performed for 12 cycles of 20 s at 98 °C and 15 s at 72 °C (14 cycles were used for the Goat Anti-Rabbit antibody due to the lower yield of input DNA). Double-size selection of libraries was performed using Agencourt AMPure XP beads (Beckman Coulter, A63880) according to the manufacturer's instruc-tions. Average library size was determined using an Agilent Tapestation DNA1000 High Sensitivity Screentape and quantification was performed using the KAPA Library Quantification Kit. CUT&RUN libraries were sequenced on a HiSeq2500 using a paired-end 125 bp run at the CRUK CI Genomics Core Facility. Reads were aligned to a reference genome containing the hg38 build of the human genome and the R64 build of the yeast genome, using subread v1.6.1[79] in paired-end genomic mode. A DB analysis was performed using the csaw package v1.14.0[80] to identify changes in histone mark enrichment in *linc00899*-depleted cells compared to the negative controls for each LOF method. Coverage was quantified by sliding a window across the genome and counting the number of sequenced fragments overlapping the window in each sample. Windows were filtered to retain only those with average abundances that were fivefold greater than the expected coverage due to background non-specific enrichment. Normalisation factors were computed by applying the TMM method[81] without weighting to the filtered windows. The fil-tered windows and normalisation factors were then used in a DB analysis with the quasi-likelihood framework in the edgeR package v3.22.0[82]. This was performed using an additive design for the generalised linear model fit, containing terms for

the batch and the depletion status (two batches in total, depletion and negative control from LNA and RNAi in each batch). A *P* value was computed for each window by first testing whether for a depletion effect in each LOF method and then taking the larger of the two *P* values[77] across both LOF methods. The above analysis was repeated using window sizes from 150 to 1000 bp to obtain DB results at varying resolutions. To consolidate these results into a single list of DB regions, overlapping windows of all sizes were clustered together based on their genomic locations, using a single-linkage approach in the consolidateWindows function. For each cluster of windows, a combined *P* value was computed using Simes' method[83]. This represents the evidence against the global null hypothesis for each cluster, i.e. that none of the constituent windows are DB. The Benjamini–Hochberg method was used to define putative DB regions at an FDR of 5%. Coverage tracks were generated by computing the number of sequencing fragments per million overlapping each based, using library sizes adjusted according to the TMM normalisation factors for each library. Tracks were visualised using the Gviz package[84].

**Generation of *C1QTNF1-AS1* CRISPR poly(A) site clones**. Insertion of the transcriptional termination signal pAS[34] into the first exon of *C1QTNF1-AS1* was performed using a published CRISPR/CAS9 protocol[85]. Briefly, the CRISPR pAS guide oligonucleotides were phosphorylated (T4 poly nucleotide kinase; NEB), annealed and ligated (Quick Ligase Kit, NEB) into *Bbs*I (NEB) digested pX458 vector (Addgene, #48138). The oligonucleotides (Sigma-Aldrich, diluted in low TE EDTA buffer at 100 μM) used for the gRNA target sequence are listed in Supplementary Table 7. All inserts were verified with Sanger sequencing. HeLa cells were plated in 6-well plates (200,000 cells/well) and, on the following day, transfected with either 2.5 μg of empty PX458 vector as a control or 2.5 μg of PX458 vector where guide oligonucleotides targeting *C1QTNF1-AS1* were cloned together with a symmetric single-stranded oligonucleotide donor[86] (ssODN; IDT Inc., 4 μl from 10 μM stock) containing the pAS flanked by homology arms (75 bp) to the target site. Lipofectamine 3000 reagent (Thermo Fisher Scientific, L3000015) was used as transfection reagent according to the manufacturer's instructions. The sequence of the symmetric ssODNs is as follows (5′ to 3′):

75 bp homology left arm-*aataaaagatctttattttcattagatctgtgtgttggttttttgtgtg*-75 bp homology right arm

*CTCACTGCGGGGCTCGGGAAGGAGGAAAGGAGTGAGCATGTCCTGCTCC
TGCATGTCCCTGCTTAAGCTCAGGA**aataaaagatctttattttcattagatctgtgtgttggtttt
ttgtgtg**CCCTTCCAGGCACCCCAGCATAGACCCCAGGACAGGGCCCCA
AGGATCCCTGGCTCATGAGAGCGGCTTGC*

The exon 1 of *C1QTNF1-AS1* is shown in upper case. The pAS site is in bold and contains *Bgl*II restriction sites (AGATCT) that were used for identification of the positive clones. The PAM site (TGG) was abrogated to avoid further cleavage by Cas9.

After 48 h, the GFP-positive cells were sorted using BD FACSAria Ilu (CRUK Flow Cytometry Core Facility) and plated on 96-well plates with DMEM supplemented with 20% FBS. Single-cell clones were expanded and genomic DNA was extracted using Direct PCR Lysis reagent (Viagen, 201-Y; 25 μl/well) with Proteinase K (Thermo Fisher Scientific, 25530-049; 0.4 mg/ml). The plates were incubated for 1 h at 55 °C, followed by 45 min incubation at 85 °C. Two μl of genomic DNA was used for screening by PCR amplification (Phusion® High-Fidelity PCR Master Mix with HF Buffer, NEB, M0531S) of the targeted genomic region. PCR conditions for *C1QTNF1-AS1*-guide 70 were: 98 °C 30 s, 98 °C 30 s, 70 °C 30 s, 72 °C 30 s (repeat 30×), 72 °C 90 s, 4 °C forever. After PCR, 5 μl was used for *Bgl*II digestion (NEB, 2 h at 37 °C) and the products were loaded on a 2% agarose gel to confirm the pAS insertion (Supplementary Fig. 8a). The rest of PCR reaction (15 μl) was also loaded on the gel as a control. Uncropped scan of the gel is provided in the Source Data file. The same genomic PCR product was also ligated into pJET1.2/Blunt and transformed into bacteria (CloneJET; Thermo Fisher Scientific, K1231). To ensure representation by both alleles, plasmids were isolated and sequenced from a minimum of 5–10 bacterial colonies. This method revealed that *C1QTNF1-AS1* pAS clones were homozygously targeted clones. Four clones were analysed (clone 95, 136, 153 and 169) by qPCR and live-cell imaging for the mitotic phenotype, along with the wild-type controls (clone 2 and clone 5) that were transfected with an empty vector PX458. We also attempted to perform pAS insertion into exon 1 of *linc00899* using two different sgRNAs. However, we failed to obtain homozygous targeted clones, most likely due to the presence of four copies of *linc00899* in HeLa cells (https://cansar.icr.ac.uk/cansar/cell-lines/HELA/copy_number_variation/). The list of PCR primers for CRISPR pAS insertion is provided in Supplementary Table 8.

**Mapping of 5′ and 3′ ends of *linc00899* and *C1QTNF1-AS1* by RACE**. HeLa total RNA (1 μg) was extracted and 5′ and 3′ RACE was performed using the Smarter RACE Kit (Clontech, 634858). cDNA was synthesised using 5′ and 3′ RACE CDS primers and SMARTer IIA oligo for template switching for 5′ RACE. cDNA ends were then amplified by touchdown PCR. The first PCR (touchdown) used the following conditions: 5 cycles of 94 °C/30 s, 72 °C/3 min; 5 cycles of 94 °C/30 s, 70 °C/30 s, 72 °C 3 min; 25 cycles of 94 °C/30 s, 68 °C/30 s, 72 °C/3 min. For nested PCR, 2 μl of PCR reaction was diluted in 98 μl Tricine-EDTA buffer and used as a template for the second PCR reaction with the following conditions: 25 cycles of 94 °C/30 s, 68 °C/30 s, 72 °C/3 min.

For *linc00899*, the first PCR used Universal Primer A mixed with *linc00899*-5R outer 1 = GATTACGCCAAGCTTacatcccggttcccacgaaaagcaacc for the 5′ ends or *linc00899*-3Routerinner3 = GATTACGCCAAGCTTccagggagggggaaaggagtcggcaat for the 3′ ends. The second (nested) PCR used Nested Universal Primer A and *linc00899*-5Routerinner1 = GATTACGCCAAGCTTggagcaggcgaagagggagtgagggg for the 5′ ends or *linc00899*-3R outerinner1 = GATTACGCCAAGCTTggtcacagc ctagccaagcccagcca for the 3′ ends.

For *C1QTNF1-AS1*, the first PCR used Universal Primer A mixed with *C1QTNF1-AS1*-5-RACE-outer-1 = GATTACGCCAAGCTTccaggcccctaatgatg tccttga for the 5′ ends or *C1QTNF1-AS1*-3RACE-outer-1 = GATTACGCCAAGC TTggaggaaaaggagtgagcatgtcctg for the 3′ ends. The second nested PCR used Nested Universal Primer A and *C1QTNF1-AS1*-5RACE-inner-2 = GATTACGCCAAGCT TGTCCTGATCTCCACCTGTCCCAAGC for the 5′ ends or *C1QTNF1-AS1*-3RACE-inner-2 = GATTACGCCAAGCTTGGAAACTTGGCAGACAGATCC AGCC for the 3′ ends.

For *linc00899*, nine different 5′ sites and four different 3′ sites were identified. For *C1QTNF1-AS1*, six different 5′ sites and four different 3′ sites were identified. The fragments were purified after agarose gel electrophoresis with the QiaQuick Gel Extraction Kit (QIAGEN), cloned in pRACE vector as per the kit instructions and transformed into Stellar component cells (Clontech, 636763). The inserts were verified by Sanger sequencing. Uncropped images of RACE results are presented in the Source Data file.

**RACE results for *linc00899***. The most common starting site for *linc00899* was CACGTCC, 87 bp upstream from the *linc00899* TSS. The most common termination site for *linc00899* was acagcaag. Capital italic letters indicate the region upstream of *linc00899* TSS (cggccgcccc) based on UCSC.

*TGGCCTCGGAGGGTCAACACCTGAGGGCCCGAGGGGCATCCATGCCC
CCTCCTCCTTCCCTCCAGCCAAAGGTGGGGGGCAAAGCGCAGGGAAGAA
AACACCAGGAGAAAACCAGAGAGCTCTCGGTTCTCTTTTAAGAGCCCTGAT
GGCCGCAGCGCAGAGCCCGAGGGGAGGGAAAATGTCGGGAAAGATTCTCTT
CCGAACTTTGCGAGTCTTTGTTTGGGAGGCTGGGGGCTGACTTCGCCGGG
GGCCGGGCCGCGGGCTCGGCCGTGCGCTCCGGTGCAGCGGCCGAGGAGCC
CCGGCGCCCGCCACCCCGGGACACGCCCTCGCAGTCGAGCCCGGACCCCGA
CCCGGACCCCAGCGCCGCCGGGCGAGGGCGGGAGGGGGAGCGCTTACCAG
ATCGTCCCGAGCGCGCCCGGGTCCAGGCGGGCACAGCGCAGGGTCAAGTT
CACGTCCGGCCCGCGGGCTGCCCGAGGTCCCCGGGCGCGGCTGGGGCAGC
GGGAGGCGCGGGAGGCCGAGGTCCGGGTGGCCGCCGcggccgccccgaagcgctg
ctgtcacccgggccgcgcccccaacttctgcacagtcgcggagctggaagtttccgggcttcgcggacacgctgggct
gggtttcagtcgcggctccgaggttggcaacaaagagggaaagaaggaggaaaagcaggccgggagaggggaa
gagaaccgcgcggaggccgcggcgccgagagccccagaacttccaattctacccagaagctttttttcgtcgtgttttt
ctcttagacatgatcctctctgaggttggtcctgggcttccatacgtgattcatggaagaggtctcagcccccaagagcccct
gagggtactgtccactcccctggaaacttccagaacctgacgtggggctgaagacatagaggctctgagagttacata
attgattctgactttggctgttggtcaacagtgtcataaggtaaaataaggctgttgtagaatctgctcagccagggagggg
aaaggagtcggcaatcaggtctcctcctgggcacctttgtgaggccagctggcagagtgggggtgacactgaggtcc
cagcagctccaaatgcaggcagagccctgtcctcagagaaggtcacagcctagccaagcccagccaggtggatgggcc
cacggaacgcacaggaacctggaacggaggttgaaagcaggaagcacagtctgtgactccccagcccactctgcattc
gaccacttggggcccagaagcttcaggaaaggtgcacaaggtcactgggtcccagtactcccaacaggaaggtctggtc
cagggacagggctcttcccgactcccctttagccacacgcaccagaagttctgcagtgcccagtgggcatagcagtccc
caagaatgacccagcactgaagctgagccaaagaacttggggacggcagccacacccctcactccctcttcgcctgctc
cagacttgccaggcggttgcttttcgtgggaaccgggatgtcctcaccaccctgtccagggcccagcccgccgatgtccc
tggcctgctacagctggaaaaaaaaaagagagatgtttgttttttatttgtttataaaaagaaaagtgtatatatataaca
tattatacctcatgaatacatacaattatttgtcaattaacaataaagaaaaatacagcaagcaaaaaagactctcttcca
caaaaatagtgttcattacagaaaagtacaaaaaaaaaaaaaaaaaaactaagaggatatttagaattaagaaaaac
taagagggtatttagaattaaaaaataaaaagaaaacaattacccatgaggtaattactgaatgcatttggttgaaa
gtccttctgctatattttccaaactgtatgtgtatatatgtgcgatgcatatatgtgttttgtgtgtgtgtgtacacatatctata
tgaatggaccatatctgtaagttataaatgcatacatatattcatgtatatataatcattagatcatactataggttattttaca
gtcttttttgttgaatacgttgtgagcatttttatgtcattattttctacaagatttgaaaaataaagtataaataccagttaa.*

Based on our sequencing results, *linc00899* transcript varied from 1144 to 1532 bp in HeLa cells. We sequenced at least 10 clones per 5′ and 3′ RACE.

**RACE results for *C1QTNF1-AS1***. The most common 5′ site was AGAGAACTA, 54 bp from the *C1QTNF1-AS1* TSS. The most common 3′ site was tctggaa. Capital italic letters indicate the region upstream of *C1QTNF1-AS1* TSS (gaaggagg) based on UCSC.

*ATCACCCCAGCACAAGTGTCACACAGCCGTGACCTTGACAAGGACCCAG
AGATAAAGATCCTTCCCACATGGCTCCGAAGCCCCTCCCTTCCTGCTCACAC
CGCATGCCTCTCCCAGAGAACTAGAGGCTGCAGCGGCAGCAGTTGGAGCA
TCTCACTGCGGGGCTCGG*gaaggaggaaaggagtgagcatgtcctgctcctggctgctaagct
caggactggccctttccaggccaaggaccccagcatagaccccaggacagggccccaaggatccctggctcatgaga
gcgcgcttgctgggctgccccaagagagcctgaaggaaacacattgttgagctgagctgacgtcgctgtttcttccagac
tgctctctaaagtgggcagggtagcgaccggccggctccgatggtgacgtcccactgccaaggggtgggagtgggga
gagtctccacagagcttcggagaagctgctaagATGGAAAAGTGGAAACTTGGCAGACA
GATCCAGCCTCCCTGGCCACTGGCCCATGCTCGTGGCTCCTGGATG
GCGCTGCCACGTTCTGAGCAGCTTGGGACAGGTGGAGATCAGGACTGG
CAGCTGCAAGGACACACCAGAGCCACAGAAACTAAAGAGAATTTCCA
AAAGGAGTCTATGGTGAAGTCTCTGAGGATGCAAAGAAGACAAGGA
GAATGAaaatccaatgaaagcctgattgtatttgttgaccttaaggaaagtgattttatggtacagcctctctg
gaagggagggtgtgttcgctcacagaatgcaaatacccttttgacccccctaatctttcttctaggagtttctcctacaga
taaacttagaaggggtgctcaaataagtaagttcaaggatatcctctgaagcattgccgtagtataaaaaaaagcacaga*

taccctcaaaggacatcattaggggcctggtaaaataaattccacacagtggaacaccgtgtagctctttagagaataaa cagctctctatatgtgatctggaaCAATCTCC.

Based on our sequencing results, *C1QTNF1-AS1* transcript varied from 864 to 952 bp in HeLa cells. We sequenced at least 10 clones per 5′ and 3′ RACE.

**Lentiviral overexpression of lncRNAs**. To produce lentivirus, HEK293T cells were first transfected with 15 µg of DNA, composed of 9 µg of the lentiviral vector DNA containing the transgene, 4 µg of psPAX.2 and 2 µg of pMD2.G in the final transfection volume of 1.5 ml (including 45 µl of Trans-Lt1 transfection reagent, Mirus) using OptiMEM medium (Thermo Fisher Scientific). Viral supernatant was collected 48 and 72 h post-transfection, spun down at $1800 \times g$ for 5 min at $+4\,°C$ and filtered through a 45-µm filter. For overexpression of *linc00899* in HeLa and RPE1 cells, *linc00899* sequence (Labomics) and negative control vector (scrambled *linc00899* sequence) were cloned into pLenti6.3/TO/V5-DEST vector (also known as lincXpress; kindly provided by John Rinn, University of Colorado) using the Gateway cloning strategy[70]. To overexpress *linc00899* from its endogenous locus using CRISPRa, we used pHAGE EF1alpha dCAS9-VP64-HA (Addgene, #50918). Two gRNA sequences targeting different regions of *linc00899* (guides 1 and 2) were cloned into pU6-sgRNA EF1Alpha-puro-T2A-BFP (Addgene, #60955). All clones were verified by Sanger sequencing using mU6 forward primers (the mU6 sequence is provided in Supplementary Table 8). HeLa and RPE1 cells were transduced with lentivirus containing dCAS9-VP64 and two gRNA targeting *linc00899* or with lentivirus contacting negative gRNA (NC2) in the presence of polybrene (5 µg/ml, Sigma). Twenty-four hours after viral transduction, the medium was exchanged and RNA was collected 48 h later for qPCR analysis. The list of CRISPRa guide sequences and primers to overexpress *linc00899* from the lincXpress vector are provided in Supplementary Tables 9 and 10, respectively.

**Immunofluorescence**. HeLa cells were seeded on coverslips in 6-well plates (120,000/well), transfected the next day with siRNA or LNA gapmers and fixed 48 h post-transfection in ice-cold methanol (Acros Organics, 167830025) for 10 min at $-20\,°C$. The cells were then washed once in $1\times$ PBS and permeabilised with PBS/0.5%Triton-X100 (Acros Organics, 327371000)/0.5%Tween-20 (Promega, H5151) for 5 min at RT followed by blocking in 5% BSA/PBS/0.001% Sodium Azide for 30 min at RT. Cells were incubated with primary antibodies overnight at 4 °C. Antibodies against acetylated α-tubulin (Sigma, T6793, dilution 1:500) and α-tubulin (Serotec, MCA78G, dilution 1:500) or against EB1 (CRUK, dilution 1:300, ref. [87]) and α-tubulin (Serotec, MCA78G, dilution 1:500) were used. Cells were washed 3× for 10 min with PBS/0.1%Tween-20 and then incubated with secondary antibodies (Supplementary Table 4) diluted in blocking-buffer for 1 h at RT. After washing again 3× for 10 min with PBS/0.1%Tween-20, cells were stained with 1 µg/ml Hoechst (B2261, Sigma, diluted in PBS) for 10 min at RT. Coverslips were mounted onto glass slides using Prolong Diamond Antifade Mountant (Thermo Fischer Scientific, P36961). To determine the mitotic index, cells were stained with antibodies against PHH3 (Millipore, 06-570, dilution 1:2000) and α-tubulin (Sigma, TUB9026, dilution 1:1000). The mitotic index was calculated by counting the cells in mitosis (positive for PHH3 and α-tubulin) and total number of cells (Hoechst positive). For each sample, at least 100 cells were randomly counted by fluorescence microscopy, and mitotic cells were scored from prophase to anaphase/telophase.

For kinetochore analysis, HeLa cells were grown on coverslips, treated with the indicated siRNAs or LNAs and fixed in PTEMF buffer (0.2% Triton X-100, 0.02 M PIPES (pH 6.8), 0.01 M EGTA, 1 mM $MgCl_2$, 4% formaldehyde) for 10 min at RT. The cells were then washed twice in $1\times$ PBS (5 min), permeabilised in 0.2% Triton X-100 in PBS for 5 min at RT, washed twice in $1\times$ PBS (5 min) and blocked as above. Cells were then incubated with primary antibodies (α-tubulin, Sigma, TUB9026, dilution 1:1000; CREST, Antibodies Inc., 15-234-0001, dilution 1:1000; Mad2, Babco, 924601, dilution 1:100) overnight at 4 °C. CREST/Mad2 images were acquired with 0.3 µm z-slices while 0.5 µm z-slices were used for acetyl α-tubulin/α-tubulin images. The list of all primary and secondary antibodies is provided in Supplementary Table 4.

**Image processing and quantification**. Imaging of fixed cells was performed on a Leica Sp8 confocal microscope using a 100 ×1.4 numerical aperture Leica oil objective. Images were taken at identical exposure times within each experiment and were imported into ImageJ and Photoshop (CS6, Adobe). Images shown here represent three-dimensional maximum intensity projections. To analyse the ratio of α-tubulin to acetylated α-tubulin, raw integrated intensities were measured (ImageJ) over the total z-stack plane using a circle selection around the mitotic spindle (based on α-tubulin). The measured intensity values were divided by the area of the selection and the background signal was subtracted. Afterwards, the mean of the signal intensity over all z-stacks was calculated, and the ratio of signal of acetylated-tubulin to α-tubulin was determined. The total EB1 levels were quantified as described above. Inter-KT distance was measured between single kinetochore pairs visible in the same z-stack plane using ImageJ. We calculated the distance between at least seven kinetochore pairs/cell from the single focal plane using the manual line measurement tool.

**Subcellular fractionation**. The cells from a 150-mm dish were used to isolate RNA from cytoplasmic, nucleoplasmic and chromatin fractions by TRIzol extraction[70]. Turbo DNA-free Kit (Thermo Fischer Scientific, AM1907) was used to remove any traces of DNA. Expression of target genes in each fraction was analysed by qPCR. Data were normalised to the geometric mean of *GAPDH* and *ACTB* levels in each cellular compartment. *U1* small nuclear RNA was used as a positive control for chromatin fraction.

**Coding potential assessment for lncRNAs**. The Coding-Potential Calculator[88] (CPC; http://cpc.cbi.pku.edu.cn) and Coding Potential Assessment Tool[89] (CPAT; http://lilab.research.bcm.edu/cpat/index.php) were used to determine noncoding potential. LncRNAs with CPC scores >1 and CPAT scores >0.364 were predicted to have protein-coding capacity. The PhyloCSF[90] score was taken from UCSC (https://github.com/mlin/PhyloCSF/wiki).

**Statistical analysis**. The statistical significance of data was determined by two-tailed Student's *t* test in all experiments using GraphPad Prism unless indicated otherwise. *P* values >0.05 were considered statistically not significant.

**Reporting summary**. Further information on research design is available in the Nature Research Reporting Summary linked to this article.

## Data availability

A reporting summary for this article is available as a Supplementary Information file. Sequencing data are available in the ArrayExpress database (http://www.ebi.ac.uk/arrayexpress) with the accession codes E-MTAB-7432 (RNA-seq), E-MTAB-7418 (CHART-seq) and E-MTAB-7419 (CUT&RUN). The imaging data have been submitted to the Image Data Resource (https://idr.openmicroscopy.org) under IDR accession number idr0056. The source data underlying Figs. 2b, d; 3a–c, e–g; 4e–i; 5c, e; 6a, i; 7a–f; 8a–c; and 9b and Supplementary Figs. 2a–d; 3a, b; 4a, b; 5a, b; 6a, c, e, f; 7a, d; 8a, c, d; 9a–c; 10a–e; and 14a, b are provided as a Source Data file. All data are available from the corresponding authors upon reasonable request.

## Code availability

All code used in this analysis is available at https://github.com/MarioniLab/LncScreen2018.

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

## Acknowledgements

We thank all the members of Gergely and Odom groups for helpful discussions and Julia Tischer for critical reading of the manuscript. We also thank the Genomics, Microscopy, FACS and Research Instrumentation and Cell Service Core Facilities at the CRUK Cambridge Institute. We thank Keith Vance (University of Bath, UK) and Matthew Simon (Yale, USA) for the help with the CHART-seq protocol, John Rinn (University of Colorado) for the lincXpress vector, Steve Henikoff (Fred Hutchinson Cancer Research Center, USA) for providing the reagents for CUT&RUN and Irene Cantone (University Federico II of Naples) for assistance with the DNA FISH. This work was made possible by funding from Cancer Research UK (C14303/A17197 to D.T.O., F.G., and J.C.M., A24455 to F.G., and A20412 to D.T.O.). We also acknowledge the support of the University of Cambridge, the Wellcome Trust (WT202878; to D.T.O.), European Research Council (615584; to D.T.O.), BBSRC Stratageic LoLa grant (BB/M00354X/1 to C.B. and A.R.B.) and Hutchison Whampoa Limited.

## Author contributions

Conception and design of study: L.S., F.G., and D.T.O. Data acquisition: L.S., A.T.L.L., P.M., C.E., A.M.R., J.M., A.R.B., and V.B. Data analyses and interpretation: L.S., A.T.L.L., P.M., A.R.B., J.C.M., C.B., D.T.O., and F.G. Writing the paper: L.S., A.T.L.L., D.T.O., and F.G. with input from all of the authors wrote the manuscript.

## Competing interests

The authors declare no competing interests.
