## [Peer Review File · Nature Communications]

Reviewers' comments:

Reviewer #1 (Remarks to the Author):

In this manuscript, Stojic et al. have screened long noncoding RNAs (lncRNAs) for function in cell division and performed more in-depth studies of one particular lncRNA called linc00899. The authors first used siRNA pools targeting about 2000 different lncRNAs and performed high content imaging to score phenotypes related to cell division. In addition to validating the top 25 hits, time-lapse microscopy was used to study cell division defects of some lncRNAs. linc00899 and lncRNA C1QTNF1-AS isoforms were characterized by RACE, and RNA FISH showed that these lncRNAs are in both the nucleus and cytoplasm. Locked nucleic acid (LNA) knockdown of these two lncRNAs had similar phenotypes as siRNA knockdown. C1QTNF1-AS was further studied with polyA signal insertion, producing a phenotype similar to transcript knockdown. The more in-depth study focused on linc00899. TPPP – a tubulin polymerization promoting protein – was increased by linc00899 knockdown, and TPPP knockdown “rescued” the cell division phenotype of linc00899 knockdown. Studies of microtubule dynamics in HeLa cells suggested that changes in TPPP levels limits the number of long-lived microtubules. Ectopic expression of linc00899 did not change TPPP levels, but CRISPR activation of the linc00899 locus did reduce TPPP levels in diploid cells. CHART-seq provided some evidence of localization of linc00899 transcript at the TPPP locus, and in RNA FISH analysis, 3% of cells had the co-localization of spliced linc00899 transcript with unspliced TPPP transcript.

Most lncRNAs have not been characterized as having important biological functions. Thus, the workflow of this screen (and identity of the screen hits) will likely be of broad interest to the lncRNA research community. The use of high-content imaging to analyze lncRNA phenotypes provides a higher resolution to the study of cell proliferation phenotypes, and, from a methodological standpoint, this aspect of the work may also be of interest. The more in-depth investigation of linc00899 function helps demonstrate the “resource-like” aspect to this manuscript. The description of isoform diversity by RACE, use of at least 2 different methods to study lncRNA depletion, and attempts at resolving cis versus trans function are commendable.

My primary comment / concern relates to the mechanistic “conclusions” drawn by the authors. The authors write, “Our mechanistic studies indicate that linc00899 regulates TPPP transcript in cis.” In the standard, commonly understood definition, cis- mechanisms take place on the same molecule of DNA. linc00899 and TPPP are on different chromosomes. There are certainly examples of lncRNAs regulating gene expression on different chromosomes by “spatial proximity,” but calling this sort of regulation “cis” might be confusing. Also, the lack of rescue by ectopic expression (which in this case was at many fold higher than normal levels) does not completely rule out potential trans function. A few suggestions follow:

Are the authors proposing that the linc00899 locus and TPPP are in close physical proximity? Have they tried DNA FISH to detect the 3D location of the linc00899 and TPPP genes? Did the authors try RNA FISH to detect unspliced, “immature” forms of linc00899 (which would be expected to be found at the linc00899 locus)? Are such immature forms of linc00899 found in the same locations as the spliced, “mature” forms of linc00899? This might be useful to investigate whether the linc00899 locus is spatially close to the TPPP locus. If these two genomic regions are close to one another, then is it possible that linc00899 knockdown causes them to become more distant?

Reviewer #2 (Remarks to the Author):

Stojic and co-authors perform a high content RNAi screen of over 2000 long noncoding RNA (lncRNAs) to identify transcripts that affect cell division. They identify two lncRNAs which alter progression through the cell cycle, linc00899 and C1QTNF1-AS1, and characterize these in more

detail. In particular, using RNA-seq, the authors identify TPPP/p25 as a differentially expressed gene after linc00899 downregulation. The authors show that loss of linc00899 leads to upregulation of the TPPP protein. TPPP is known to inhibit the histone deacetylase HDAC6, a well characterized regulator of tubulin acetylation, and in line with this increased tubulin acetylation is observed in linc00899 depleted cells. The authors also analyse the mechanisms by which linc00899 affects TPPP transcription and conclude that regulation of the TPPP locus most likely involves linc00899 likely binding to the TPPP locus via protein interactions rather than direct binding to the DNA, although the precise nature of this is not defined.

Regulation of mitotic progression by lncRNAs is an underexplored area and the manuscript is therefore timely and will be of interest to a wide readership. The experiments presented are of high quality and generally support the claims made in the text. However, the cell biology and precise nature of the mitotic defects could be explored more.

Specific comments:

- The cell biological defects in mitotic progression observed with the two lncRNAs are not characterised very well. The images in the manuscript are very small and the temporal resolution of the live cell imaging is not great. It is therefore difficult for the reader to discern what is actually wrong with the progression through the cell cycle in the affected cells. It would be helpful to have additional stainings with kinetochore and additional spindle markers to visualise defects more clearly.
- The authors initially focus on two lncRNAs that they identify in their screen, linc00899 and C1QTNF1-AS1. When they perform RNA-seq only for linc00899 differentially expressed target genes are identified. In contrast, for C1QTNF1-AS1 only C1QTNF1-AS1 itself comes up as a differentially expressed gene when C1QTNF1-AS1 is depleted. What does this mean for the mitotic function of C1QTNF1-AS1? The interpretation of these results is not well explained in the text.
- Depletion of linc00899 results in a prominent cell cycle delay at the metaphase-to-anaphase transition with an aligned metaphase spindle. This is quite an unusual arrest phenotype and looks reminiscent of an inhibition of the anaphase promoting complex. How does this fit with the known role of TPPP/p25 in cell cycle progression?
- Figure 4B shows that TPPP/p25 is upregulated upon linc00899 RNAi whereas ITGB1BP1, Ral14 and DNAAF5 are downregulated. Only TPPP/p25 is then further characterised. What about the other genes? How do they contribute to the mitotic phenotype observed upon linc00899 RNAi?

Response to Reviewers, October 2019.

Stojic et al., “A high-content RNAi screen reveals multiple roles for long noncoding RNAs in cell division”.

Considered for publication in *Nature Communications*

We thank the Reviewers for their constructive feedback that has significantly improved our manuscript. We have performed new experiments to address their comments and marked the changes in the revised manuscript by red text.

Reviewer 1

Reviewers' comments:

In this manuscript, Stojic et al. have screened long noncoding RNAs (lncRNAs) for function in cell division and performed more in-depth studies of one particular lncRNA called linc00899. The authors first used siRNA pools targeting about 2000 different lncRNAs and performed high content imaging to score phenotypes related to cell division. In addition to validating the top 25 hits, time-lapse microscopy was used to study cell division defects of some lncRNAs. linc00899 and lncRNA C1QTNF1-AS isoforms were characterized by RACE, and RNA FISH showed that these lncRNAs are in both the nucleus and cytoplasm. Locked nucleic acid (LNA) knockdown of these two lncRNAs had similar phenotypes as siRNA knockdown. C1QTNF1-AS was further studied with polyA signal insertion, producing a phenotype similar to transcript knockdown. The more in-depth study focused on linc00899. TPPP – a tubulin polymerization promoting protein – was increased by linc00899 knockdown, and TPPP knockdown “rescued” the cell division phenotype of linc00899 knockdown. Studies of microtubule dynamics in HeLa cells suggested that changes in TPPP levels limits the number of long-lived microtubules. Ectopic expression of linc00899 did not change TPPP levels, but CRISPR activation of the linc00899 locus did reduce TPPP levels in diploid cells. CHART-seq provided some evidence of localization of linc00899 transcript at the TPPP locus, and in RNA FISH analysis, 3% of cells had the co-localization of spliced linc00899 transcript with unspliced TPPP transcript.

Most lncRNAs have not been characterized as having important biological functions. Thus, the workflow of this screen (and identity of the screen hits) will likely be of broad interest to the lncRNA research community. The use of high-content imaging to analyze lncRNA phenotypes provides a higher resolution to the study of cell proliferation phenotypes, and, from a methodological standpoint, this aspect of the work may also be of interest. The more in-depth investigation of linc00899 function helps demonstrate the “resource-like” aspect to this manuscript. The description of isoform diversity by RACE, use of at least 2 different methods to study lncRNA depletion, and attempts at resolving cis versus trans function are commendable.

We thank the Reviewer for her/his positive comments.

My primary comment / concern relates to the mechanistic “conclusions” drawn by the authors. The authors write, “Our mechanistic studies indicate that linc00899 regulates TPPP transcript in cis.” In the standard, commonly understood definition, cis-mechanisms take place on the same molecule of DNA. linc00899 and TPPP are on different chromosomes. There are certainly examples of lncRNAs regulating gene expression on different chromosomes by “spatial proximity,” but calling this sort of regulation “cis” might be confusing. Also, the lack of rescue by ectopic expression (which in this case was at many fold higher than normal levels) does not completely rule out potential trans function.

In response to these comments, we have amended the relevant sections in both the Results and Discussion (page 15, 16 and 18).

A few suggestions follow:

Are the authors proposing that the *linc00899* locus and *TPPP* are in close physical proximity? Have they tried DNA FISH to detect the 3D location of the *linc00899* and *TPPP* genes?

To address if *linc00899* regulation of *TPPP* expression involves spatial proximity, as suggested by the Reviewer, we performed 3D DNA FISH (adapted from Edith Heard lab) in HeLa and RPE1 cells. Unfortunately, the high background of *TPPP* BAC probes (red) in both cell types and the low expression of *TPPP* in the diploid RPE1 cell line prevented us from obtaining publication-worthy conclusive results (Figure 1 a-c). Our preliminary data however reveals colocalization (white arrow, Figure 1 a-b) of *linc00899* and *TPPP* at least in a few cells. We attach a number of representative confocal images for *linc00899* (green) and *TPPP* (red) in both cell lines. We have also toned down our conclusions and revised the text to accurately reflect our data (please see page 15, line 25; page 16, line 5-15; page 18 line 24 for text).

a

DNA FISH, *exp1*, *Linc00899* green; *TPPP* red (BAC Chori *TPPP*; BAC Invitrogen *linc00899*)

b

DNA FISH, *exp2*, *Linc00899* green; *TPPP* red (BAC Chori *TPPP*; BAC Invitrogen *linc00899*)

c

Figure 1. a, b) DNA FISH for *linc00899* (green) and *TPPP* (red) in HeLa and RPE1 cells, Scale bar, 20 μ m. **c)** Relative expression of *linc00899* and *TPPP* in HeLa and RPE1 cells (n=11 replicates).

Did the authors try RNA FISH to detect unspliced, “immature” forms of *linc00899* (which would be expected to be found at the *linc00899* locus)? Are such immature forms of *linc00899* found in the same locations as the spliced, “mature” forms of *linc00899*? This might be useful to investigate whether the *linc00899* locus is spatially close to the TPPP locus. If these two genomic regions are close to one another, then is it possible that *linc00899* knockdown causes them to become more distant?

We have tried to detect unspliced form of *linc00899* using intronic probes. We have designed two sets of intronic probes against intron 1 and intron 3 but found that only the probe against the first intron (intron 1) worked for RNA FISH. Although, in some cells we observed the immature form of *linc00899* to be at the same location as the spliced form of *linc00899* (exonic probes), we cannot exclude the possibility that the intronic probe (intron 1) also detects certain spliced isoforms of *linc00899*. Indeed, several *linc00899* isoforms containing intron 1 have been reported in different lncRNA annotations (*Cabili et al., Genes Dev, 2011; Hezroni et al., Cell Reports, 2015*) (Figure 2 a, b).

Figure 2. a, b) UCSC snapshot for *linc00899* isoforms based on *Cabili et al., 2011* and *Hezroni et al., 2015*. The Stellaris RNA FISH probes against intron 1 and intron 3 of *linc00899* are indicated.

Reviewer 2

Reviewer #2 (Remarks to the Author):

Stojic and co-authors perform a high content RNAi screen of over 2000 long noncoding RNA (lncRNAs) to identify transcripts that affect cell division. They identify two lncRNAs which alter progression through the cell cycle, *linc00899* and *C1QTNF1-AS1*, and characterize these in more detail. In particular, using RNA-seq, the authors identify TPPP/p25 as a differentially expressed gene after *linc00899* downregulation. The authors show that loss of *linc00899* leads to upregulation of the TPPP protein. TPPP is known to inhibit the histone deacetylase HDAC6, a well characterized regulator of tubulin acetylation, and in line with this increased tubulin acetylation is observed in *linc00899* depleted cells. The authors also analyse the mechanisms by which *linc00899* affects TPPP transcription and conclude that regulation of the TPPP locus most likely involves *linc00899* likely binding to the TPPP locus via protein interactions rather than direct binding to the DNA, although the precise nature of this is not defined. Regulation of mitotic progression by lncRNAs is an underexplored area and the manuscript is therefore timely and will be of interest to a wide readership. The experiments presented are of high quality and generally support the claims made in the text. However, the cell biology and precise nature of the mitotic defects could be explored more.

We thank the Reviewer for her/his positive comments.

Specific comments:

The cell biological defects in mitotic progression observed with the two lncRNAs are not characterised very well. The images in the manuscript are very small and the temporal resolution of the live cell imaging is not great. It is therefore difficult for the reader to discern what is actually wrong with the progression through the cell cycle in the affected cells. It would be helpful to have additional stainings with kinetochore and additional spindle markers to visualise defects more clearly.

To provide greater insight into the mitotic progression defects, we have now characterised the loss-of-function phenotypes for *linc00899* and *C1QTNF1-AS1* in more depth. We have performed additional experiments and now include high resolution confocal micrographs of the mitotic phenotypes.

As the Reviewer suggested, we have now performed new immunofluorescence staining for CREST (kinetochore marker) and alpha-tubulin (microtubules). These experiments allowed us to characterise mitotic phenotypes in detail after depletion of *linc00899* and *C1QTNF1-AS1* lncRNAs. We found that depletion of the two lncRNAs led to chromosome congression defects. Depletion of *linc00899* did not preclude formation of metaphase plates, but individual chromosomes were frequently found near the poles. By contrast, *C1QTNF1-AS1*-depleted cells exhibited a more severe phenotype with chromosomes failing to congress to a metaphase plate, resulting in groups of unattached chromosomes (new Figure 5f and new Supplementary Figure 11; please see page 10, line 11 and page 13, line 23 for text).

To assess whether the level of tubulin acetylation observed in *linc00899*-depleted cells would be sufficient to trigger defects in microtubule dynamics, we compared it to tubulin acetylation in taxol treated cells. We used low dose of taxol, which is known to suppress microtubule dynamics (without affecting spindle morphology) and increase acetylated tubulin levels in the spindle (*Jordan et al., Cancer Res, 1996; Derry et al., Biochemistry, 1995; Jordan et al., PNAS 1993*). We compared cells treated with 0.5-3nM taxol to those depleted of *linc00899*, and found tubulin acetylation in *linc00899*-depleted cells to be similar to those treated with 3nM taxol (new Figure 5a-c; please see page 12, line 29 for text). These new data are now replacing the previous images and corresponding quantifications.

The authors initially focus on two lncRNAs that they identify in their screen, *linc00899* and *C1QTNF1-AS1*. When they perform RNA-seq only for *linc00899* differentially expressed target genes are identified. In contrast, for *C1QTNF1-AS1* only *C1QTNF1-*

AS1 itself comes up as a differentially expressed gene when C1QTNF1-AS1 is depleted. What does this mean for the mitotic function of C1QTNF1-AS1? The interpretation of these results is not well explained in the text.

As we did not observe changes in gene expression after *C1QTNF1-AS1* depletion by RNA-seq, these data suggest that *C1QTNF1-AS1* is unlikely to act as a transcriptional regulator. *C1QTNF1-AS1* could of course influence translation of mRNAs, but we favour the possibility that *C1QTNF1-AS1* regulates cell division through its protein interactors (please see page 11, line 25 for text). Indeed, several lncRNAs have been shown to modulate enzymatic activities of proteins (lncRNA *Inc-DC*, Wang *et al.*, *Science* 2014; lncRNA *CONCR*, Marchese *et al.*, *Mol Cell* 2016) where functionality of *C1QTNF1-AS1* could depend on its ability to act as a cofactor of its protein partner (Marchese *et al.*, *Genome Biol*, 2017).

Depletion of linc00899 results in a prominent cell cycle delay at the metaphase-to-anaphase transition with an aligned metaphase spindle. This is quite an unusual arrest phenotype and looks reminiscent of an inhibition of the anaphase promoting complex. How does this fit with the known role of TPPP/p25 in cell cycle progression?

Prompted by this comment, we now include detailed analysis of the different mitotic spindle phenotypes observed in lncRNA-depleted cells. We noted congression defects upon depletion of both lncRNAs. In case of *linc00899*, single chromosome pairs were present near the poles in ~20% of cells, and the majority of these exhibited Mad2 signal, suggesting intact SAC signalling in these cells (new Figure 5g). Furthermore, by measuring interkinetochore distances, we found these to be reduced in *linc00899*-depleted cells, which is consistent with abnormal microtubule dynamics (new Figure 5h-j; please see page 14, line 8-30 for text). Our results therefore support a SAC-dependent mitotic delay rather than an arrest due to a failure to activate the APC/C.

Figure 4B shows that TPPP/p25 is upregulated upon linc00899 RNAi whereas ITGB1BP1, Rai14 and DNAAF5 are downregulated. Only TPPP/p25 is then further characterised. What about the other genes? How do they contribute to the mitotic phenotype observed upon linc00899 RNAi?

We have included the mRNA expression and protein levels of other *linc00899* targets identified by RNA-seq. Our new data suggest that depletion of *linc00899* did not trigger consistent and significant changes in the levels of RAI14, ITGB1BP1 and DNAAF5 with RNAi and LNA Gapmers (new Supplementary Figure 9; please see page 11, line 29 for text). Based on these data we decided not to perform further functional assays.

REVIEWERS' COMMENTS:

Reviewer #1 (Remarks to the Author):

In this revision, Stojic and colleagues have performed new experiments and revised the text to address the comments of the reviewers. In response to my own primary comment/concern, the authors have amended the relevant sections in the Results/Discussion. These revisions are appropriate, but I have just a few very minor comments that the authors may consider addressing with minor text revisions:

1. The authors say that they use CRISPRa to "assess a possible cis effect." I still find the use of the genetic term "cis" to be potentially confusing. It might be more accurate to say that CRISPRa tests the effect of linc00899 activation in the context of its normal genomic context. For instance, when produced from the genome, linc00899 undergoes splicing, which might be important to its function.
2. The "title" of Fig. 7, "Overexpression of linc00899 has differential effect on TPPP expression," doesn't quite reflect the data in the figure (there is both OE and KD data). Maybe just, "Gain-of-function and rescue studies of linc00899," would work?

The authors also made laudable attempts at testing my suggestion that linc00899 and TPPP might be in close physical proximity, but these results (shown as reviewer figure 1 and 2) were not "publication-worthy," which I find understandable. I appreciate their efforts.

In summary, I believe that this manuscript is suitable for publication and will be of broad interest to the lncRNA community.

Reviewer #2 (Remarks to the Author):

The authors added a significant amount of extra data to the manuscript, which has been strongly improved. All my concerns have been addressed. I have just one comment: in Figure 3, in the live cell imaging of C1QTNF1-AS1 depleted or C1QTNF1-AS1 LNA1 treated cells, the spindle poles are much more visible than in the other live cell imaging panels. It is a bit difficult to discern whether this is an imaging effect (better focus in these sets of images than for the other movies) or a biological effect. I wonder whether the authors could either exchange the current stills for an alternative set - in case of a technical reason for the enhanced spindle poles) or at least comment on this phenomenon in the text?

Point-by-point Response to Reviewers, December 2019.

Stojic et al., “A high-content RNAi screen reveals multiple roles for long noncoding RNAs in cell division”.

Considered for publication in *Nature Communications*

Again, we would like to thank the Reviewers for their constructive criticism. We have now amended the text and addressed their additional comments. All the changes in the revised manuscript are tracked and are in red.

Reviewer 1

Reviewers' comments:

In this revision, Stojic and colleagues have performed new experiments and revised the text to address the comments of the reviewers. In response to my own primary comment/concern, the authors have amended the relevant sections in the Results/Discussion. These revisions are appropriate, but I have just a few very minor comments that the authors may consider addressing with minor text revisions:

1. The authors say that they use CRISPRa to “assess a possible cis effect.” I still find the use of the genetic term “cis” to be potentially confusing. It might be more accurate to say that CRISPRa tests the effect of linc00899 activation in the context of its normal genomic context. For instance, when produced from the genome, linc00899 undergoes splicing, which might be important to its function.

We have now corrected the corresponding text on page 15, line 31.

2. The "title" of Fig. 7, "Overexpression of linc00899 has differential effect on TPPP expression," doesn't quite reflect the data in the figure (there is both OE and KD data). Maybe just, “Gain-of-function and rescue studies of linc00899,” would work?

We have changed the title of the Figure 7 (new Figure 8) to: “Gain-of-function and rescue studies of linc00899 and its effect on TPPP expression” (page 66, line 6).

The authors also made laudable attempts at testing my suggestion that linc00899 and TPPP might be in close physical proximity, but these results (shown as reviewer figure 1 and 2) were not “publication-worthy,” which I find understandable. I appreciate their efforts.

We thank the Reviewer for noting our attempts regarding the DNA FISH.

In summary, I believe that this manuscript is suitable for publication and will be of broad interest to the lncRNA community.

We thank the Reviewer for her/his positive evaluation of our work.

Reviewer 2

Reviewers' comments:

The authors added a significant amount of extra data to the manuscript, which has been strongly improved. All my concerns have been addressed.

We are pleased to read that the Reviewer is now satisfied with our revised version of the manuscript.

I have just one comment: in Figure 3, in the live cell imaging of C1QTNF1-AS1 depleted or C1QTNF1-AS1 LNA1 treated cells, the spindle poles are much more visible than in the other live cell imaging panels. It is a bit difficult to discern whether this is an imaging effect (better focus in these sets of images than for the other movies) or a biological effect. I wonder whether the authors could either exchange the current stills for an alternative set - in case of a technical reason for the enhanced spindle poles) or at least comment on this phenomenon in the text?

We apologise for not observing this effect earlier. We went through our live-cell movies again and identified examples with a better overall focus and exchanged these in Figure 3h. The new panels in Figure 3h for *linc00899*-depleted cells using LNAs are now more comparable in terms of focus and this should now help the readers appreciate the mitotic phenotype better. We believe that the apparent increase in α -tubulin intensities at the poles in *C1QTNF1-AS1* depleted cells is most likely due to shorter and sparser spindles, especially when the spindles become multipolar. We have also included a short sentence describing this in the manuscript (page 10, line 24).